# Gut microbiota-bile acid crosstalk contributes to the rebound weight gain after calorie restriction in mice

Mengci Li [1,2], Shouli Wang[2], Yitao Li[3], Mingliang Zhao[1,2], Junliang Kuang[2], Dandan Liang[2], Jieyi Wang[2], Meilin Wei[2], Cynthia Rajani[4], Xinran Ma[5], Yajun Tang[2], Zhenxing Ren[2], Tianlu Chen[2], Aihua Zhao[2], Cheng Hu[6], Chengxing Shen[7], Weiping Jia [6], Ping Liu[1], Xiaojiao Zheng [2✉] & Wei Jia [2,3✉]

Calorie restriction (CR) and fasting are common approaches to weight reduction, but the maintenance is difficult after resuming food consumption. Meanwhile, the gut microbiome associated with energy harvest alters dramatically in response to nutrient deprivation. Here, we reported that CR and high-fat diet (HFD) both remodeled the gut microbiota with similar microbial composition, *Parabacteroides distasonis* was most significantly decreased after CR or HFD. CR altered microbiota and reprogramed metabolism, resulting in a distinct serum bile acid profile characterized by depleting the proportion of non-12α-hydroxylated bile acids, ursodeoxycholic acid and lithocholic acid. Downregulation of UCP1 expression in brown adipose tissue and decreased serum GLP-1 were observed in the weight-rebound mice. Moreover, treatment with *Parabacteroides distasonis* or non-12α-hydroxylated bile acids ameliorated weight regain via increased thermogenesis. Our results highlighted the gut microbiota-bile acid crosstalk in rebound weight gain and *Parabacteroides distasonis* as a potential probiotic to prevent rapid post-CR weight gain.

[1] School of Biomedical Engineering and Med-X Research Institute, Shanghai Jiao Tong University, Shanghai 200030, China. [2] Center for Translational Medicine and Shanghai Key Laboratory of Diabetes Mellitus, Shanghai Jiao Tong University Affiliated Sixth People's Hospital, Shanghai 200233, China. [3] School of Chinese Medicine, Hong Kong Baptist University, Kowloon Tong, Hong Kong 999077, China. [4] University of Hawaii Cancer Center, Honolulu, HI 96813, USA. [5] Shanghai Key Laboratory of Regulatory Biology, Institute of Biomedical Sciences and School of Life Sciences, East China Normal University, Shanghai 200062, China. [6] Department of Endocrinology and Metabolism, Shanghai Jiao Tong University Affiliated Sixth People's Hospital, Shanghai Diabetes Institute, Shanghai 200233, China. [7] Department of Cardiology, Shanghai Jiao Tong University Affiliated Sixth People's Hospital, Shanghai 200233, China. ✉email: joyzheng99@sjtu.edu.cn; weijia1@hkbu.edu.hk

Obesity is a chronic disease that has reached pandemic proportions and leads to a number of metabolic diseases such as type 2 diabetes (T2DM), cardiovascular diseases, and cancer. Anti-obesity approaches, such as caloric restriction, aerobic-exercise, pharmacological treatment or bariatric surgery, lead to weight loss and reduce disease risks in individuals with obesity[1–3]. The calorie restriction (CR) diet has been recognized as a factor that promotes health, extends longevity, and prevents the development of metabolic and age-related diseases[4]. However, long-term weight-loss maintenance has proven to be challenging due to complex interactions between hormones and behavior[5,6] which often leads to a gradual or rapid weight regain after CR intervention[7,8]. Studies have suggested that CR as a dietary intervention reshaped the gut microbiota composition, causing changes in the host metabolism[9]. The gut microbiota under CR is deprived of 40–60% of the nutrients and thus, will shift towards favoring bacteria that can more efficiently harvest energy and depleting organisms that are less efficient. In fact, one of the obesity theories is the overabundance of highly efficient energy-harvesting bacteria[10]. In human studies, CR decreased abundances of *Clostridium perfringens*, *Ruminococcus gnavus*, and *Akkermansia muciniphila* and increased relative abundances of probiotic microbes, such as *Bifidobacterium spp*[9]. In a long-term (141 weeks) murine study, 141-week of CR decreased the abundances of the operation taxonomic units (OTUs) in Lachnospiraceae and *Parabacteroides*[11].

The crosstalk between gut microbiota and host metabolism has been reported in many studies, and among the extracellular metabolites derived from the host, bile acids (BAs) are a highly abundant pool of host-derived and microbial-modified metabolites that are major regulators of glucose and lipid homeostasis[12–14]. Primary BAs are synthesized in hepatocytes, conjugated with glycine or taurine, and are released into the intestine to facilitate the absorption of nutritional ingredients. In the intestine, gut microbes modify BAs structure and function. For example, conjugated BAs can be deconjugated through microbial bile salt hydrolase (BSH) activity and primary BAs such as chenodeoxycholic acid (CDCA) can be transformed by microbial 7α-hydroxysteroid dehydrogenase (7α-HSDH) into secondary BAs such as ursodeoxycholic acid (UDCA), or by 7α-dehydroxylase to form lithocholic acid (LCA)[12]. Takeda G protein-coupled receptor 5 (TGR5) and farnesoid X receptor (FXR) are two major BA receptors that regulate lipid and glucose homeostasis[15]. Both FXR and TGR5 regulate glucagon-like peptide-1 (GLP-1) levels and modulate glucose metabolism[16]. Our group has recently reported that increased non-12α-hydroxylated BAs (non-12OH BAs) lead to an obesity-resistant phenotype in mice through TGR5 mediated brown adipose tissue (BAT) activation and upregulation of uncoupling protein 1 (UCP1) expression[17].

In this study, we focused on the mechanism of weight regain after CR. We found that CR reshaped the gut microbial ecosystem by producing an elevated ratio of Firmicutes to Bacteroidetes, a microbiota compositional change similar to that observed in mice exposed to a high-fat diet (HFD). In particular, *Parabacteroides distasonis* was significantly reduced after host exposure to CR or HFD. *Parabacteroides distasonis* is capable of producing secondary BAs. We found the proportion of non-12OH secondary BAs, such as UDCA and LCA, was significantly decreased in the CR group. Upon the reinstatement of food intake, the abundance of bacteria for efficient energy harvest and 12α-hydroxylated BAs (12OH BAs) for efficient fat absorption increased significantly, resulting in weight rebound. Supplementation with *Parabacteroides distasonis* or non-12OH BAs in mice ameliorated weight regain via increasing the proportion of non-12OH BAs and promoting thermogenesis.

## Results

### The reinstatement of chow diet or HFD after fasting and CR resulted in rapid weight gain and obesogenic metabolic phenotypes.

To determine whether recovery from fasting (food withdrawal) or CR would lead to weight gain, we designed two experiments. Fasting involves refraining from food consumption for a period of time. The beneficial effects of fasting include reduced body weight, delayed aging, and improved health[18,19]. Meanwhile, there have been some undesirable outcomes reported for fasting such as weight rebound or the development of food intolerance and inflammation[20,21]. Therefore, we designed this "fasting" paradigm in mice that begins with stepwise initiation (50% CR for 4 days), the final status of fasting (4 days of 100% CR), and gradual recovery of food consumption (50% CR for 4 days) by lean or obese mice. The outline of the fasting experiment is diagrammed in Fig. 1a. Half of the mice in each group were euthanized before fasting while the remaining mice had a period of fasting followed by a 4-week recovery with access to chow diet ad libitum before the final time point. The body weight was decreased during fasting and then recovered in 4 weeks (Fig. 1b). The oral glucose tolerance test (OGTT) was performed before fasting and after the recovery. The area under the curve (AUC) of OGTT was increased after the 4-week recovery on the chow diet and reached the level of HFD, implying impaired glucose tolerance (Fig. 1c). The fasting blood glucose elevated after the 4-week recovery in both chow diet and HFD (Fig. 1d). Liver weight and the liver weight index (liver weight/body weight) were significantly increased after the 4-week recovery, especially for those mice on the chow diet (Fig. 1e and Supplementary Fig. 1a). We found that fasting caused a more significant liver index rebound in the chow diet group relative to the HFD group after the 4-week recovery. We measured the body composition before and after the fasting, we found that fat mass was significantly reduced after fasting in two kinds of diets (Supplementary Fig. 1b).

For further examination of the risk of weight regain after CR diet, we conducted a second experiment. Male C57BL/6J mice in CD, HF, and CR groups were fed with chow diet, HFD, and CR diet respectively for 12 weeks and then switched to HFD (Fig. 1f). Body weight of the mice in the CR + HF group increased at a rapid rate in the follow-up period of HFD (Fig. 1g). The average energy intake was calculated based on the daily food intake of each mouse and the average energy intake in the CR + HF group was significantly less than the CD + HF group during the HFD (Supplementary Fig. 1c). No significant changes were found in the excretion of energy in feces (Supplementary Fig. 1d). Thus, after the same amount of energy intake the mice on CR diet had the most weight gain (Supplementary Fig. 1e). Consistent with the nocturnal lifestyle of mice, all groups showed higher energy expenditure during the night than during the light. The energy expenditure of CR + HF mice was significantly lower than that of control mice in the dark stage, which was supported the hypothesis that weight regain occurs by decreasing energy expenditure (Fig. 1h). The decreasing energy expenditure was found to be independent of physical activity (analysis of variance (ANOVA), $p = 0.39$, Supplementary Fig. 1f).

Regarding blood glucose, the CR + HF group mice showed impaired glucose tolerance (Fig. 1i) and affected insulin resistance (Fig. 1j) compared with the CD + HF group mice. During the follow-up of excessive energy intake after CR, the weights of liver and epididymal white adipose tissue (WAT) were elevated (Fig. 1k). Meanwhile, the weights of BAT showed a significant increase in both CD + HF and HF + HF groups but only a marginal change in the CR + HF group (Fig. 1k). The elevated liver index (Fig. 1l) indicated hepatic steatosis, which was verified by hematoxylin and eosin (H&E) staining of the liver tissue

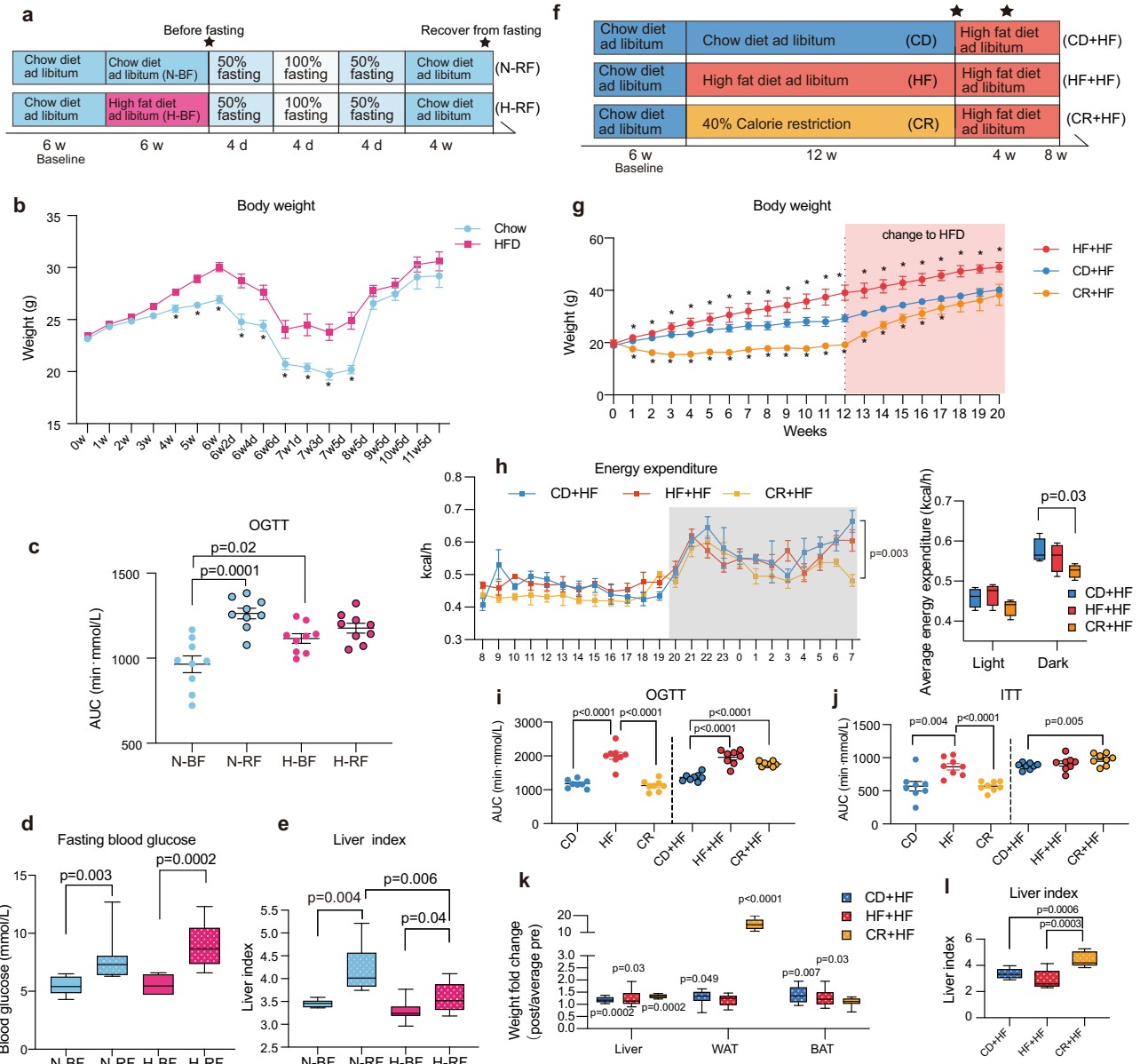

**Fig. 1 The reinstatement of chow diet or HFD after fasting and CR resulted in rapid weight gain and obesogenic metabolic phenotypes. a** The experimental workflow of the fasting experiment. The stars indicated the time of the samples collection. **b** Body weight in the fasting experiment. (*$p < 0.05$) **c** The AUC of OGTT. **d** Fasting blood glucose level. **e** Liver index level. **f** Workflow of the diets changing experiment. The stars indicated the time of the samples collection. **g** Body weight during the diets changing experiment ($n = 8$ per group in the 16-week experiment and $n = 6$ per group for the observation of weight regain in 20 weeks. Differences were compared with the CD + HF group, *$p < 0.05$). **h** Raw energy expenditures and raw average energy expenditures in the period of light and dark, $n = 4$ per group. The hypothesis tests result of analysis-of-covariance with timely body mass as a covariate was shown in the left panel. **i** AUC of the OGTT. **j** AUC of the ITT. **k** Liver, WAT, and BAT weights fold change after change to HFD, the differences were compared between the pre and post changing diets. **l** Liver index level. $n = 9$ per group in the fasting experiment, $n = 8$ per group in the diets changing experiment. Data are expressed as means ± SEM (**c**, **i**, **j**). $p$-values in figures were calculated by the two-tailed unpaired T-test in the GraphPad software. All box and whiskers plots showed the box (from the 25th to 75th percentiles), the median value (in the transverse line), and the whiskers (go down to the smallest value and up to the largest). Source data are provided as a Source data file. Oral glucose tolerance test (OGTT); insulin tolerance tests (ITT); area under the curve (AUC); high-fat diet (HFD); epididymal white adipose tissue (WAT); brown adipose tissue (BAT).

(Supplementary Fig. 1g) from the CR + HF group. After the HFD was reinstated, the healthy liver condition caused by CR changed to fat accumulation rapidly. We also found adipocyte hypertrophy in WAT of the CR + HF group (Supplementary Fig. 1g). The rapid weight rebound along with increased blood glucose and fat accumulation in liver and adipose tissue was observed in the post-CR recovery period, suggesting the obesogenic metabolic phenotype.

**Remodeling of gut microbiota under CR or HFD**. We suspected that CR facilitated natural selection for organisms with higher energy-harvesting capability along with the decreased abundance of less efficient organisms, thus leading to a bodyweight regain upon diet reinstatement. We collected the cecal contents from the mice in HF, CD, and CR groups and performed whole-genome shotgun metagenomic sequencing using the Illumina platform (Supplementary Fig. 2a). Principal coordinate analysis (PCoA) at

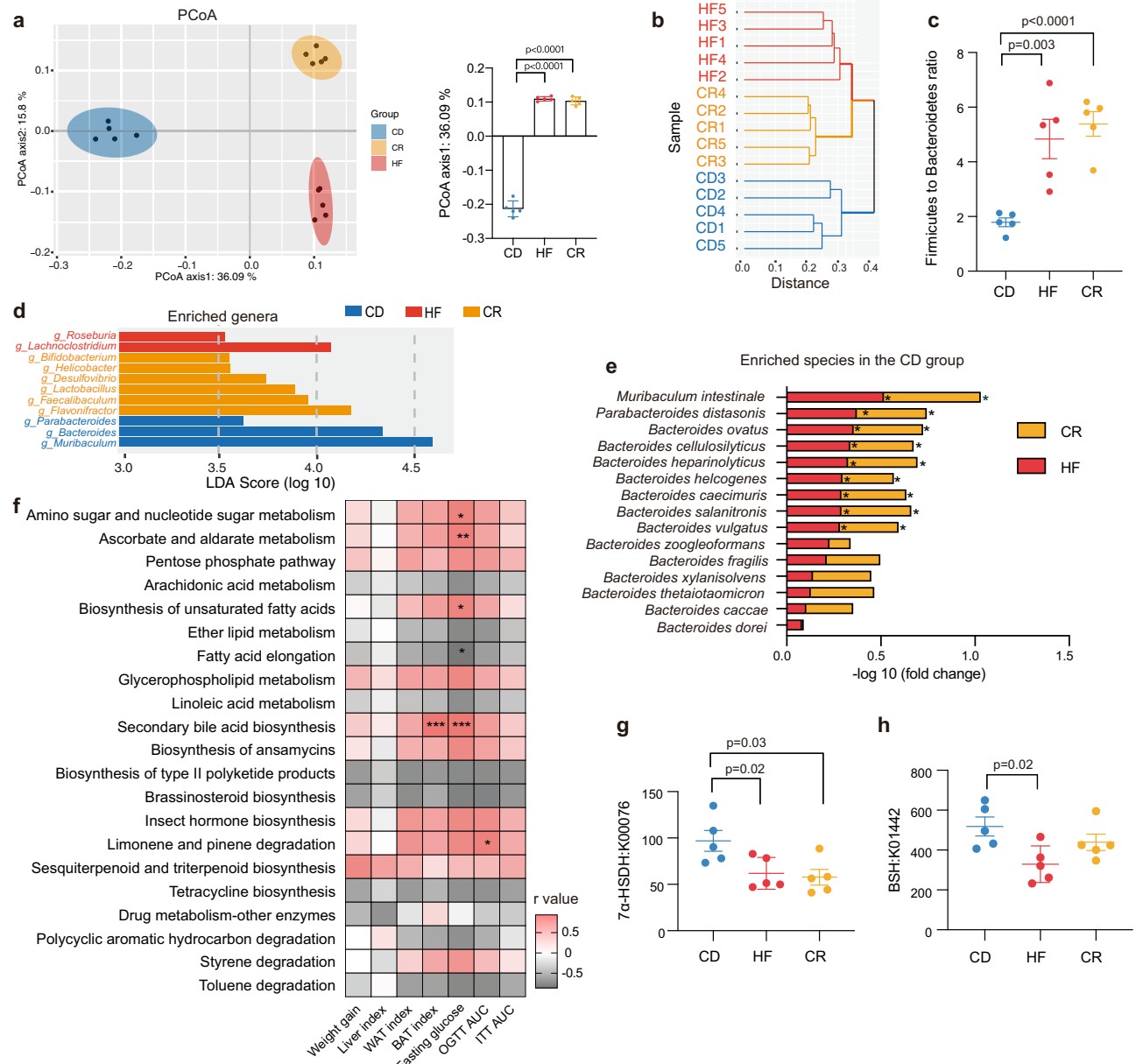

**Fig. 2 Remodeling of gut microbiota under CR or HFD. a** PCoA plot based on the Canberra similarity of cecal microbial composition. PCoA1 axis level of the three groups. **b** Hierarchical clustering based on the Canberra similarity among three groups. **c** The ratio of Firmicutes to Bacteroidetes among three groups. **d** LDA effect size method was performed to compare enriched taxa (levels of genus) in each group. The bar plot listed the significantly differential taxa (score >3.5) in HF, CR, and CD groups. **e** Fold change of the species enriched in the CD group with the most abundance. **f** Heatmap of correlation coefficients between the KEGG L3 pathway and phenotypes of the individuals with similar body weight before 4-week HFD by Spearman correlation. Each correlation coefficient smaller than −0.8 or bigger than 0.8 was included in the plot, the p-values were adjusted by the method of Bonferroni. The color of each spot in the heatmap corresponds to the r value. **g** Relative abundance of the 7α-HSDH (K00076) pathway. **h** Relative abundance of the BSH (K01442) pathway. n = 5 per group. Data are expressed as mean ± SEM in the bar plots. All p-values in figures were calculated by the two-tailed unpaired T-test in the GraphPad software. Source data are provided as a Source data file. Oral glucose tolerance test (OGTT); insulin tolerance tests (ITT); area under the curve (AUC); epididymal white adipose tissue (WAT); brown adipose tissue (BAT); bile salt hydrolase (BSH); 7α-hydroxysteroid dehydrogenase (7α-HSDH); principal coordinate analysis (PCoA); linear discriminant analysis (LDA).

the species level showed that the microbiota composition was significantly different among the three groups (Fig. 2a and Supplementary Fig. 2b) and that the PCoA axis 1 of both HF and CR groups showed similar levels relative to the CD group. Hierarchical clustering was implemented at the species level, and we found that CR and HF groups clustered together more closely than the CD group (Fig. 2b). At the phylum level, we found that Firmicutes increased and Bacteroidetes decreased in the HF as well as in the CR group. The relative abundance of Actinobacteria

was elevated in both CR and CD groups (Supplementary Fig. 2c). The ratio of Firmicutes to Bacteroidetes was significantly elevated in both HF and CR groups (Fig. 2c). Differences in the taxa composition of the three groups were assessed using linear discriminant analysis (LDA) effect size (Supplementary Fig. 2d and Supplementary Fig. 2e). We found that the families of Bifidobacteriaceae and Eggerthellaceae from the Actinobacteria phylum were enriched in CR and CD groups, respectively. Although the Firmicutes phylum abundance was elevated in both CR and HF

groups, the specific elevated classes were disparate: *Lactobacillus* and *Faecalibaculum* genera from Bacilli and Erysipelotrichia classes were enriched in the CR group, *Lachnoclostridium* and *Roseburia* genera from the Clostridia class were enriched in the HF group. We also found some taxa from the Bacteroidetes class that was decreased in the CR and HF groups, such as *Bacteroides* and *Parabacteroides* (Fig. 2d). Then we analyzed the profile of high abundance species from these enriched genera (Supplementary Fig. 2f). Some common species from the *Bacteroides* and *Parabacteroides* genera were significantly decreased in both HF and CR groups (Fig. 2e). We then conducted the Kyoto Encyclopedia of Genes and Genomes (KEGG) pathway analysis of the microbiome data hierarchically. In the highest level of the KEGG Orthology (L1), the "Metabolism" pathway accounted for the highest proportion (Supplementary Fig. 2g) of the diet-affected pathways. Then we performed the partial least-squares discriminant analysis (PLS-DA) of the second level (L2) in the "Metabolism" pathway (Supplementary Fig. 2h), HF and CR groups showed less distance from one another than from the CD group. Four significantly altered L2 pathways were selected for further analysis, including "Carbohydrate metabolism", "Metabolism of terpenoids and polyketides", "Lipid metabolism", and "Xenobiotics biodegradation and metabolism" (Supplementary Fig. 2i). Among them, the third level (L3) pathway of "Secondary bile acid biosynthesis" showed the strongest correlation with the most phenotypes (Fig. 2f). The abundance of 7α-HSDH pathway was significantly lowered in the CR group compared with the CD group (Fig. 2g), and the pathway of BSH was also showed a decreasing trend (Fig. 2h).

**The depletion of *Parabacteroides distasonis* and non-12OH BAs in CR.** We measured the BA profiles of the cecal contents, liver, and serum in the CD, HF, and CR groups using ultra-performance liquid chromatography-triple-quadrupole mass spectrometry (UPLC/TQ-MS, ACQUITY UPLC, Waters Corp., Milford, MA, USA) and observed distinct clusters of BA profiles among the three groups of mice in the PLS-DA scores plot (Supplementary Fig. 3a). The concentration of total BAs was fluctuated significantly among the three groups as assessed by the Kruskal–Wallis test (Fig. 3a). We found that both the CR and HF mice showed elevated level of 12OH BAs and a lowered proportion of non-12OH BAs in their BA profiles, especially those that were unconjugated (Fig. 3b). In serum, the proportion of unconjugated 12OH BAs including cholic acid (CA) and deoxycholic acid (DCA) was increased in the CR group mice and unconjugated non-12OH BAs including LCA and UDCA were significantly decreased (Fig. 3c, d; the relevant absolute values were shown in Supplementary Fig. 3b). The conjugated non-12OH BAs, including tauro-β-muricholic acid (TβMCA), tauro-ω-muricholic acid (TωMCA), tauroursodeoxycholic acid (TUDCA), and taurochenodeoxycholic acid (TCDCA), were decreased in the CR group (Supplementary Fig. 3c). In the cecal contents, unconjugated non-12OH BAs, especially UDCA, were significantly reduced in both the CR and HF groups (Fig. 3e and Supplementary Fig. 3d). Using generalized correlation analysis for metabolome and microbiome (GRaMM)[22], we found that *Parabacteroides distasonis* had a strong correlation with most unconjugated non-12OH BAs (Fig. 3f), including UDCA, LCA, and muricholic acid (MCA) species (α-muricholic acid (αMCA) and β-muricholic acid (βMCA)).

In the liver, unconjugated non-12OH BAs were significantly reduced in the CR group while conjugated 12OH BAs were elevated in both CR and HF groups (Fig. 3b). To explore the possible cause of the differences observed in the liver BA profiles, we performed mRNA analyses of liver tissues to evaluate the expression of BA synthesis enzymes. FXR and its downstream molecule small heterodimer partner (SHP) are known to inhibit hepatic cytochrome P-450 cholesterol 7α-hydroxylase (CYP7A1) expression[12]. HF and CR diets were both shown to inhibit the expression of *Shp* in liver (Supplementary Fig. 3e). The mRNA expression level of *Cyp7a1* was significantly elevated in the CR group compared with the CD group, which was consistent with the result of elevated concentration of total BAs. The protein expression of oxysterol 7α-hydroxylase (CYP7B1), a key enzyme in the alternate pathway of BA synthesis and was mainly responsible for the synthesis of non-12OH BAs, was significantly lower in the CR group (Supplementary Fig. 3f).

**The depletion of *Parabacteroides distasonis* and non-12OH BAs after fasting.** To visualize the microbiome remodeling after fasting and a period of recovery, a PLS-DA of OTUs was performed, and the results were shown in Fig. 4a. Before fasting, the chow diet (N-BF) and HFD (H-BF) groups were separated from each other. After fasting and a period of recovery, the two groups of chow diet-recover from fasting (N-RF) and HFD-recover from fasting (H-RF) were clustered towards H-BF, especially in axis 1 (Supplementary Fig. 4a). Then we performed the PLS-DA analysis of all species between N-BF and N-RF groups and between H-BF and H-RF groups (Supplementary Fig. 4b and Supplementary Fig. 4c). The variable important in projection (VIP) scores greater than 1.5 from the PLS-DA analysis were exhibited in Fig. 4b and Supplementary Fig. 4d. *Parabacteroides distasonis* was the key gut microbe that resulted in the group separation before and after fasting. The relative abundance of *Parabacteroides distasonis* was significantly decreased after fasting on a chow diet (Fig. 4c). Consistent with the previous experiment, the proportion of non-12OH BAs was decreased after diet restriction (Fig. 4d). The taurine-conjugated 12OH BAs were significantly increased after fasting in both chow diet and HFD groups (Fig. 4e), while the proportion of secondary unconjugated non-12OH BA, especially UDCA, was significantly decreased after fasting in the chow diet (Fig. 4f, the relevant absolute values were shown in Supplementary Fig. 4e).

**The depletion of *Parabacteroides distasonis* and non-12OH BAs resulted in reduced energy expenditure in the weight-rebound mice.** We compared the body weight of germ-free mice that received the gut microbiome from the donors in the diets changing experiment and found that the relative weight gain was significantly increased in the mice of R-trans and H-trans groups (Supplementary Fig. 4f). In the diets changing experiment, the energy expenditures of the CR + HF group mice were attenuated (Fig. 1h). Then we examined the relative abundance of *Parabacteroides distasonis* in the cecal contents by PCR and a persistent decrease was observed in the CR + HF group (Fig. 5a). The measured BA levels in cecal contents, liver and serum revealed that the concentration of total BAs were significantly altered among the three groups (Kruskal–Wallis test, $p < 0.05$) (Supplementary Fig. 5a). In hierarchical clustering analysis of the BA profiles in serum after changing to HFD (Supplementary Fig. 5b), we found the profiles of the CR + HF group were more similar to those of the HF + HF group than the CD + HF group. Relative proportions of non-12OH BAs and unconjugated non-12OH BAs were significantly decreased in the CR + HF group (Fig. 5b and Supplementary Fig. 5c). A reduced proportion of unconjugated non-12OH BAs was detected in the CR + HF group, including MCAs (Fig. 5c), LCA, UDCA (Fig. 5d), and the taurine-conjugated non-12OH BA, TβMCA. The relevant absolute values were shown in Supplementary Fig. 5d. On the other hand, the secondary 12OH BAs, including DCA and taurodeoxycholic acid (TDCA), were significantly increased in the CR + HF or HF + HF group (Supplementary Fig. 5e).

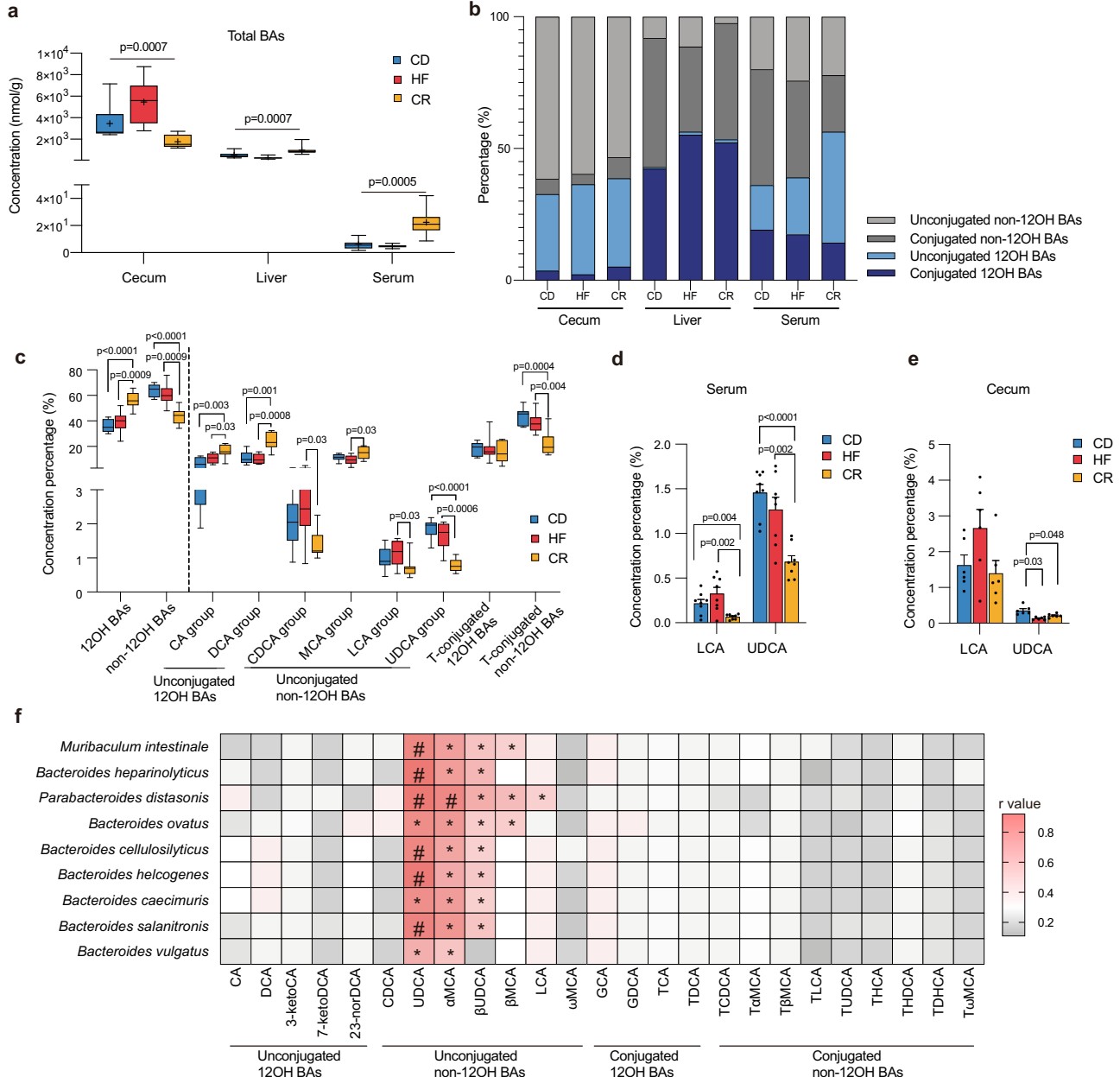

**Fig. 3 The depletion of *Parabacteroides distasonis* and non-12OH BAs in CR. a** Concentration of total BAs in contents of cecum, liver, and serum among three groups. Three groups variation was assessed by the Kruskal–Wallis test. **b** The bar plots show the mean percentage of (non-)12OH and (un) conjugated BAs in contents of cecum, liver, and serum among three groups. **c** The BAs concentration percentage profiles in the serum of three groups. **d** Dysregulated non-12OH unconjugated BAs (LCA and UDCA) composition in the serum of three groups. **e** Dysregulated non-12OH unconjugated BAs (LCA and UDCA) composition in the cecal contents of three groups. **f** Heatmap of GRaMM's correlation coefficients between the cecal BAs and the species significantly enriched in the CD group, the color of each spot in the heatmap corresponds to the *r* value, *$p < 0.05$; #$p < 0.0001$. $n = 8$ per group. Differences in the BAs concentration percentage data were assessed by the two-tailed multiple T-test in the GraphPad software, *p*-values were adjusted by the FDR's method. Data are expressed as mean ± SEM in (**d**, **e**). All box and whiskers plots showed the box (from the 25th to 75th percentiles), the median value (in the transverse line), and the whiskers (go down to the smallest value and up to the largest). Source data are provided as a Source data file. 12α-hydroxylated bile acids (12OH BAs); non-12α-hydroxylated bile acids (non-12OH BAs); T-conjugated (taurine-conjugated); cholic acid (CA); deoxycholic acid (DCA); chenodeoxycholic acid (CDCA); muricholic acid (MCA); lithocholic acid (LCA); ursodeoxycholic acid (UDCA); 3-ketocholic acid (3-ketoCA); 7-ketodexycholic acid (7-ketoDCA); 23-nordeoxycholic acid (23-norDCA); α-muricholic acid (αMCA); 3β-ursodeoxycholic acid (βUDCA); β-muricholic acid (βMCA); ω-muricholic acid (ωMCA); glycocholic acid (GCA); glycodehydrocholic acid (GDCA); taurocholic acid (TCA); taurodeoxycholic acid (TDCA); taurochenodeoxycholic acid (TCDCA); tauro α-muricholic acid (TαMCA); tauro β-muricholic acid (TβMCA); taurolithocholic acid (TLCA); tauroursodeoxycholic acid (TUDCA); taurohyocholic acid (THCA); taurohyodeoxycholic acid (THDCA); taurodehydrocholic acid (TDHCA); tauro ω-muricholic acid (TωMCA).

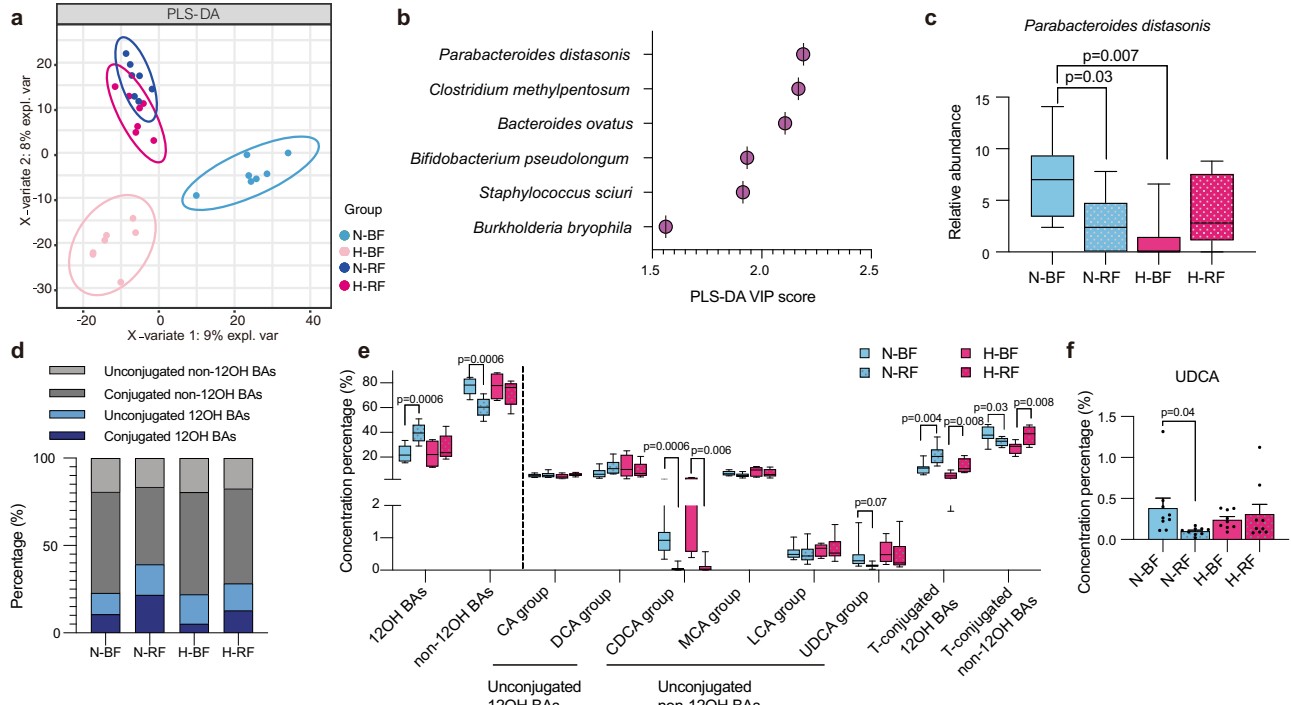

**Fig. 4 The depletion of *Parabacteroides distasonis* and non-12OH BAs after fasting. a** PLS-DA plot of the cecal microbiome in four groups, $n = 7$ per group. **b** The species rank of VIP scores of the PLS-DA analysis in chow diet. **c** Relative abundance of *Parabacteroides distasonis* among four groups. **d** The bar plots of the mean percentage of (non-)12OH and (un)conjugated BAs in the serum. **e** The BAs composition profiles in serum. *p*-values were adjusted by the FDR's method. **f** UDCA was significantly reduced after fasting in the chow diet, data are expressed as means ± SEM. $n = 9$ per group. Differences were assessed by two-tailed multiple T-test in the GraphPad software. All box and whiskers plots showed the box (from the 25th to 75th percentiles), the median value (in the transverse line), and the whiskers (go down to the smallest value and up to the largest). Source data are provided as a Source data file. Partial least-squares discriminant analysis (PLS-DA); variable important in projection (VIP); cholic acid (CA); deoxycholic acid (DCA); chenodeoxycholic acid (CDCA); muricholic acid (MCA); lithocholic acid (LCA); ursodeoxycholic acid (UDCA); 12α-hydroxylated bile acids (12OH BAs); non-12α-hydroxylated bile acids (non-12OH BAs); T-conjugated (taurine-conjugated).

We then hypothesized that the altered BA profile seen in the CR + HF group had a negative effect on the overall homeostasis of the mice. The mRNA expression in the liver of *Shp*, fibroblast growth factor receptor 4 (*Fgfr4*), *Fxr*, and BA synthetases were measured. The results showed the same tendency as previously determined for *Cyp7b1* and the *Fxr* pathway-related gene mRNA expressions (Supplementary Fig. 5f and Supplementary Fig. 5g). LCA was one of the downregulated unconjugated non-12OH BAs detected in the CR + HF group and acted as the strongest TGR5 agonist. We measured the *Tgr5* and *Gcg* (precursor of GLP-1) mRNA expression levels in the ileum and found them to be decreased in the CR + HF group (Fig. 5e), which suppressed the serum level of active GLP-1 (Fig. 5f). A GLP-1 receptor agonist could rescue the weight regain in CR + HF mice (Supplementary Fig. 5h). *Tgr5* is also expressed in BAT, which regulates the expenditure of energy. In BAT, the mRNA and protein expression of UCP1 was markedly downregulated in mice from HF + HF and CR + HF groups (Fig. 5g, h). Peroxisome proliferator-activated receptor γ coactivator-1 α (*Pgc1α*) mRNA expression was showed a decreasing trend in the HF + HF group but was not significant in the CR + HF group (Fig. 5g). Elongation of very long-chain fatty acids 6 (*Elovl6*), which is necessary for the thermogenic action of BAT, showed significantly reduced expression in both HF + HF and CR + HF groups (Fig. 5g).

**The treatment with *Parabacteroides distasonis* attenuated weight rebound after CR diet.** To investigate the beneficial effects of *Parabacteroides distasonis*, we fed the CR + HF mice

with *Parabacteroides distasonis* (CR + HF + PD group), vehicle (CR + HF + Vehicle group), or heat-killed *Parabacteroides distasonis* (CR + HF + HKPD group) by oral gavage daily for 4 weeks while the mice were on HFD (Supplementary Fig. 6a). The weight gain of the CR + HF + PD group was significantly reduced compared with the CR + HF + Vehicle or CR + HF + HKPD groups (Fig. 6a). The weight rebound showed no significant change between the *Parabacteroides distasonis* supplementation and CD + HF group (Supplementary Fig. 6b). The average energy intake of each mouse at the first week of recovery (Fig. 6b) and the fecal energy excretion among the three groups showed no significant change (Supplementary Fig. 6c). The intervention of *Parabacteroides distasonis* increased the energy expenditure in the light and dark period (Fig. 6c and Supplementary Fig. 6d), and the physical activity showed no significant difference (ANOVA, $p = 0.19$, Supplementary Fig. 6e). Moreover, we found the beneficial effects such as decreased levels of fasting blood glucose (Fig. 6d), improved glucose tolerance and insulin sensitivity (Supplementary Fig. 6f), decreased weights of the fat mass (Fig. 6e), and several tissues (Supplementary Fig. 6g) in the *Parabacteroides distasonis* supplementation group.

Profiling of the cecal microbiota composition using 16S rRNA gene sequencing showed a similar microbial composition between the Vehicle and HKPD groups. The microbiota of the CR + HF + PD group was distinct from the other two groups (Fig. 6f). The mean abundance of the Firmicutes was decreased and Bacteroidetes elevated after the administration of *Parabacteroides distasonis* (Fig. 6g). In the intestine and serum, *Parabacteroides distasonis* treatment significantly increased the proportion of the unconjugated

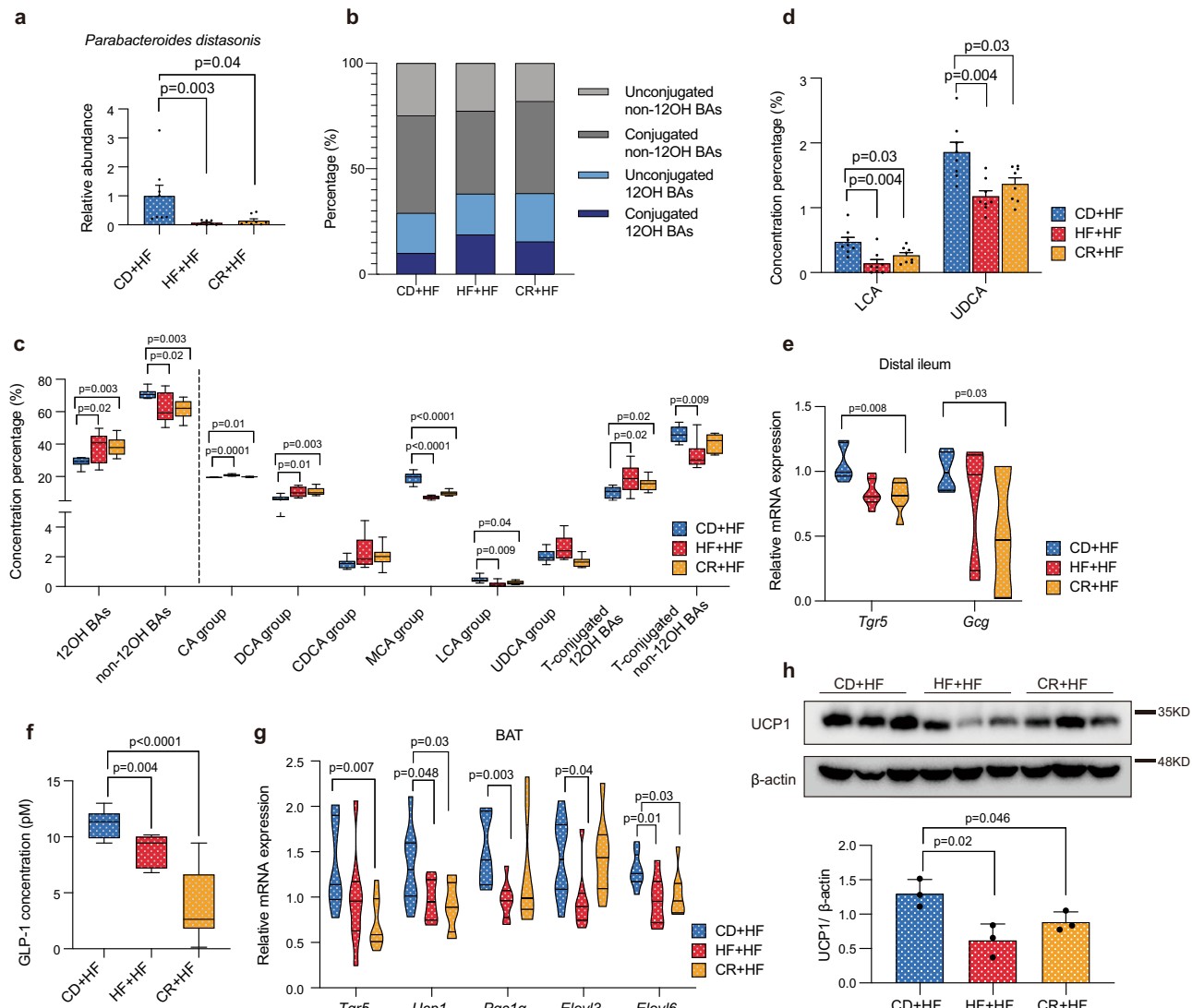

**Fig. 5 The depletion of *Parabacteroides distasonis* and non-12OH BAs resulted in reduced energy expenditure in the weight-rebound mice. a** Relative abundance of *Parabacteroides distasonis* in the contents of cecum among three groups. **b** The (un)conjugated (non-)12OH BAs percentage in the serum. **c** The BAs composition profiles in the serum. *p*-values were adjusted by the FDR's method. **d** The composition level of LCA and UDCA in the serum. *p*-values were adjusted by the FDR's method. **e** Relative mRNA expression of *Tgr5* and *Gcg* genes in the distal ileum, *n* = 6 per group. **f** Active GLP-1 level in serum. **g** Relative mRNA expression of thermogenesis-related genes in the BAT. **h** UCP1 protein expression in the BAT, *n* = 3 per group. *n* = 8 per group in the experiments. Differences were assessed by a two-tailed multiple T-test in the GraphPad software. Data were expressed as means ± SEM in the bar plots. All box and whiskers plots showed the box (from the 25th to 75th percentiles), the median value (in the transverse line), and the whiskers (go down to the smallest value and up to the largest). Source data are provided as a Source data file. 12α-hydroxylated bile acids (12OH BAs); non-12α-hydroxylated bile acids (non-12OH BAs); T-conjugated (taurine-conjugated); cholic acid (CA); deoxycholic acid (DCA); chenodeoxycholic acid (CDCA); muricholic acid (MCA); lithocholic acid (LCA); ursodeoxycholic acid (UDCA); Takeda G protein-coupled receptor 5 (*Tgr5*); glucagon-like peptide-1 (GLP-1); brown adipose tissue (BAT); uncoupling protein 1 (*Ucp1*); peroxisome proliferator-activated receptor-γ coactivator-1α (*Pgc1α*); elongation of very long-chain fatty acids 3 (*Elovl3*); elongation of very long-chain fatty acids 6 (*Elovl6*).

non-12OH BAs (Fig. 6h and Supplementary Fig. 6h), especially UDCA and LCA (Fig. 6i, j). In addition, glycoursodeoxycholic acid (GUDCA), taurolithocholic acid (TLCA), and TβMCA were also increased in the CR + HF + PD group (Supplementary Fig. 6i). Therefore, *Parabacteroides distasonis* supplementation was able to restore the BA profile from CR + HF condition. The mRNA expression of *Gcg* (Fig. 6k) and the active GLP-1 level (Fig. 6l) showed significant upregulation in the CR + HF + PD group, close to the level of the CD + HF group (Fig. 5f). The expression of UCP1 was significantly elevated in BAT (Fig. 6m, n). In UCP1-knockout (UKO) mice, the intervention of *Parabacteroides distasonis* shifted the microbial metabolites as the same way in the wild-type mice

(Supplementary Fig. 6j) and elevated the proportion of non-12OH BAs (Supplementary Fig. 6k and Supplementary Fig. 6l) but could not rescue the weight regain (Fig. 6o), as well as the glucose homeostasis (Fig. 6p and Supplementary Fig. 6m). As a result, our data suggested that *Parabacteroides distasonis* treatment mainly increased the levels of the unconjugated non-12OH BAs, UDCA, and LCA, which acted as agonists to activate the GLP-1 and UCP1 pathway and improve glucose and energy metabolism.

**The increased non-12OH BA attenuated the weight regain after CR diet.** To further investigate the roles of non-12OH BAs

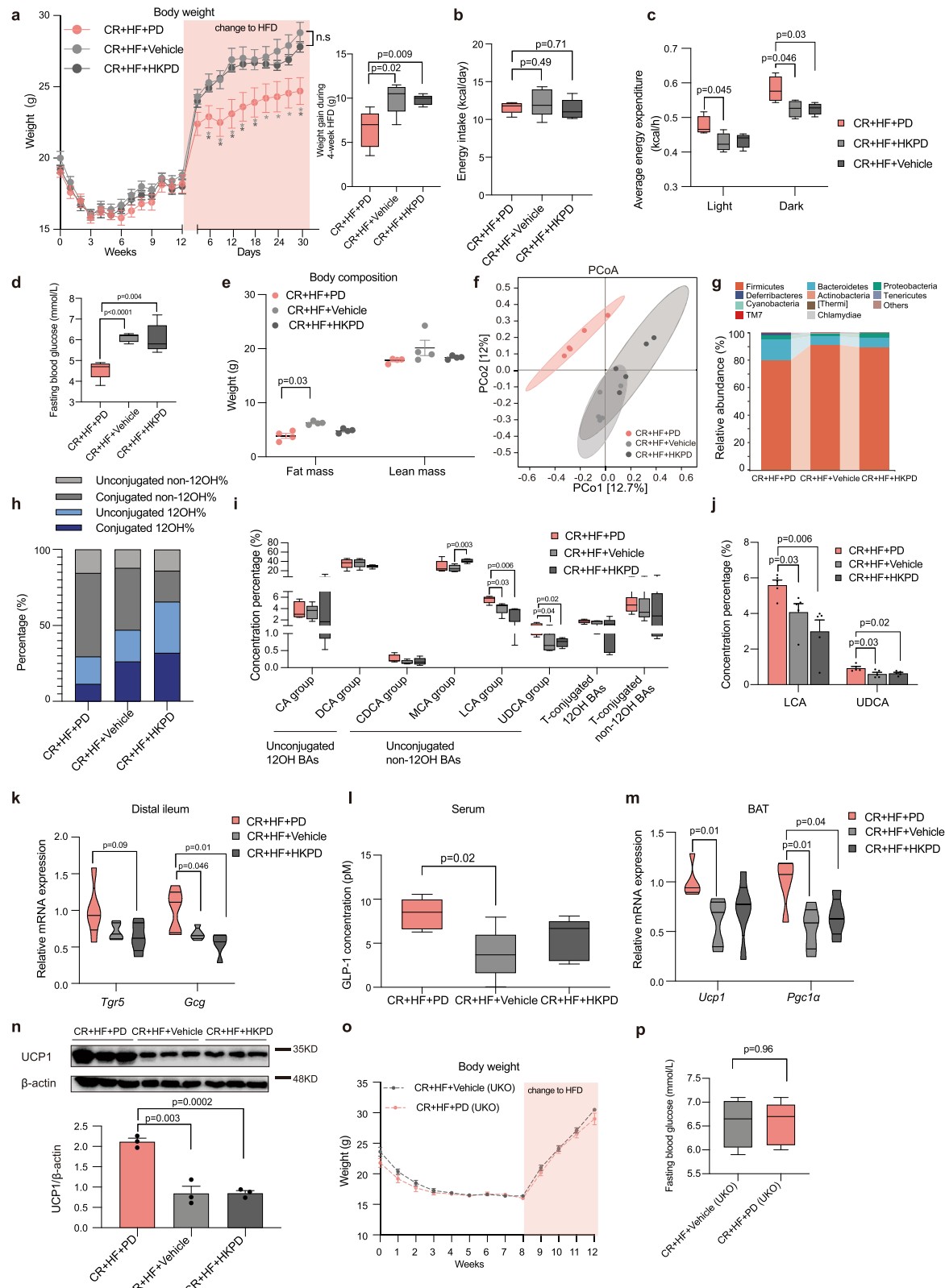

in obesity development, we performed an experiment utilizing HFD mice supplemented with 0.5% (w/w) UDCA after CR (Supplementary Fig. 7a). During HFD, the weight gain of the UDCA treatment group was significantly decreased (Fig. 7a), while the average energy intakes at the first week of the recovery (Fig. 7b) and the fecal energy excretion (Supplementary Fig. 7b) showed no significant differences between CR + HF + UDCA

and the control group. The administration of the non-12OH BA increased the energy expenditure in the mice (Supplementary Fig. 7c), which was independent of the physical activity (ANOVA, $p = 0.17$, Supplementary Fig. 7d). Fasting blood glucose, and the AUC of OGTT and ITT showed a significant reduction in the CR + HF + UDCA group (Fig. 7c, d), as well as decreased weights of fat mass and several tissues (Fig. 7e and Supplementary

**Fig. 6 The treatment with *Parabacteroides distasonis* attenuated weight rebound after CR diet. a** Body weight (*$p < 0.05$. Differences between the CR + HF + PD group and the CR + HF + Vehicle group are marked in light-gray asterisks, differences between the CR + HF + PD group and the CR + HF + HKPD group are marked in dark-gray asterisks.) and the weight gain during the 4-week HFD. **b** Average energy intake during the first week of changing to HFD. **c** Raw average energy expenditures per hour during the period of light and dark, $n = 4$ per group. **d** Fasting blood glucose at the end of the experiment. **e** Fat mass and lean mass of the mice by NMR miniSpec LF50, $n = 4$ per group. **f** PCoA plots of the cecal microbiome based on the Jaccard similarity. **g** Mean abundance of the phyla among the three groups. **h** The bar plots show the mean percentage of (non-)12OH and (un)conjugated BAs in the serum. **i** The BAs composition profiles in the cecal contents among three groups. **j** The composition of unconjugated non-12OH BAs (LCA and UDCA). $p$-values were adjusted by the FDR's method. **k** Relative mRNA expression of *Tgr5* and *Gcg* in the distal ileum. **l** Active GLP-1 level in serum. **m** Relative mRNA expression of *Ucp1* and *Pgc1α* in BAT. **n** UCP1 protein expression in BAT. **o** The body weight of UKO mice. **p** The fasting blood glucose level of UKO mice. $n = 5$ in the CR + HF + PD, CR + HF + Vehicle, CR + HF + HKPD, and CR + HF + PD (UKO) groups. $n = 4$ in the CR + HF + Vehicle (UKO) group. All differences were assessed by the two-tailed multiple T-test in the GraphPad software. All box and whiskers plots showed the box (from the 25th to 75th percentiles), the median value (in the transverse line), and the whiskers (go down to the smallest value and up to the largest), data in bar plots were expressed as means ± SEM. Source data are provided as a Source data file. High-fat diet (HFD); principal coordinate analysis (PCoA); 12α-hydroxylated bile acids (12OH BAs); non-12α-hydroxylated bile acids (non-12OH BAs); T-conjugated (taurine-conjugated); cholic acid (CA); deoxycholic acid (DCA); chenodeoxycholic acid (CDCA); muricholic acid (MCA); lithocholic acid (LCA); ursodeoxycholic acid (UDCA); bile acid (BA); Takeda G protein-coupled receptor 5 (*Tgr5*); glucagon-like peptide-1 (GLP-1); uncoupling protein 1 (*Ucp1*); peroxisome proliferator-activated receptor-γ coactivator-1α (*Pgc1α*); brown adipose tissue (BAT).

Fig. 7e). The total BAs concentration was significantly increased in the serum of the UDCA treatment mice (Supplementary Fig. 7f). In the CR + HF + UDCA group, the abundance of unconjugated non-12OH BAs markedly increased (Fig. 7f), especially UDCA and LCA (Fig. 7g). We then measured the expression of BA synthetases in the liver, to further investigate the origin of BA composition remodeling. *Cyp7b1* was significantly increased after UDCA treatment. Meanwhile, sterol-12α-hydroxylase (*Cyp8b1*) was significantly decreased at the mRNA level (Fig. 7h). Along with the increase of non-12OH BAs (LCA and UDCA), the expression of UCP1 was increased (Fig. 7i and Supplementary Fig. 7g), however, the increase in *Pgc1α* mRNA expression was not significant ($p = 0.057$). In a UKO model, elevated non-12OH BAs (Supplementary Fig. 6k and 6l) could not significantly reduce the body weight (Fig. 7j), fasting glucose (Fig. 7k), and the AUC of ITT test in CR + HF mice (Supplementary Fig. 6m). We found the AUC of OGTT decreased in the CR + HF + UDCA (UKO) mice, implying the influence of BAs on other pathways of blood glucose regulation. Therefore, the increased synthesis of non-12OH BAs attenuated the weight regain after CR mediated by UCP1-related pathways.

## Discussion

Weight loss and related beneficial effects of calorie restriction have been shown in many studies[23,24]. However, long-term weight-loss maintenance is challenging. Here we simulated fasting followed by chow diet recovery and the extreme condition of CR followed by transfer to a period of HFD in mice and showed that resuming high-calorie diet-induced weight gain and loss of metabolic improvement due to the weight loss. Gut microbes play an important role in this stage. We determined that the bacterium, *Parabacteroides distasonis* was significantly reduced during fasting or CR. After the reinstatement of a regular diet or HFD, there was a rapid weight gain with increased metabolic dysfunction. The supplementation of *Parabacteroides distasonis* and the non-12OH BA, UDCA to the HFD prevented weight gain and improved the glucose and energy metabolism.

There is evidence that suggests a correlation between the intestinal bacteria composition and weight loss caused by CR diet or weight gain caused by HFD, however, the results regarding CR-induced alterations in the relative abundances of main phyla bacteria have varied including Firmicutes, Bacteroidetes, and Proteobacteria[25–28]. Antje et al. investigated gut microbial changes and energy harvest induced by laparoscopic sleeve gastrectomy (LSG) and a CR diet in subjects with obesity. They

found that LSG, but not CR, improved the obesity-associated gut microbiota composition (Firmicutes to Bacteroidetes ratio) towards a lean microbiome phenotype[25]. Beate et al. found some OTUs belonging to the family Lachnospiraceae within the phylum Firmicutes that showed higher relative abundance after CR in women with obesity[28]. Here we also found that CR reduced the Bacteroidetes population and in favor of Firmicutes. HFD-induced obesity is correlated with an increased ratio of Firmicutes to Bacteroidetes[29], and here we demonstrated that both the CR and HFD increased the ratio of Firmicutes to Bacteroidetes with some variations in genera and species. Some species that were decreased in both CR and HF compared with chow diet, included *Parabacteroides distasonis*, *Bacteroides ovatus*, *Bacteroides cellulosilyticus*, etc. We found that some of them had a BSH function, but it showed no significant differences in KEGG pathway analysis (Fig. 2h). *Bacteroides ovatus* has been reported as the species that best-induced gut IgA production[17]. *Parabacteroides distasonis* is one of the 14 core microbes in the gut of human beings and has been proven to transform primary to secondary BAs to produce DCA, UDCA, and LCA, in vitro[30]. In another study, colonization with *Parabacteroides distasonis* resulted in very low amounts of DCA[31]. *Parabacteroides distasonis* has been shown to meet some of the requirements for a probiotic key microorganism[32] including more abundance in healthy persons than in the diseased ones, ability to isolate and culture the strain as a pure culture in vitro and ability to mitigate the disease through transplant or other forms of supplementation. Our results showed that administration of *Parabacteroides distasonis* as a supplement remodeled the BA profile to elevate levels of the non-12OH BAs, LCA and UDCA. On the other hand, our data showed that CR induced enrichment of some gut microbes such as *Lactobacillus johnsonii* and *Bifidobacterium pseudolongum*. Charlet et al. showed that *Lactobacillus johnsonii* and *Bifidobacterium pseudolongum* enhanced the anti-inflammatory cytokine response with high Toll-like receptor 9 expression and attenuated the development of colitis in mice[33]. We further performed all KEGG pathway functional analyses. Differential enriched pathways compared with the CD group were identified by LDA effect size (Supplementary Fig. 2j). The pathways enriched in both the HF and CR groups included ATP-binding cassette (ABC) transporter, cell motility, pentose phosphate metabolism, carbohydrate metabolism, bacterial chemotaxis, amino sugar, and nucleotide sugar metabolism. Glycerolipid metabolism and pyruvate metabolism pathways were enriched in the CR group, which suggested that structural and functional

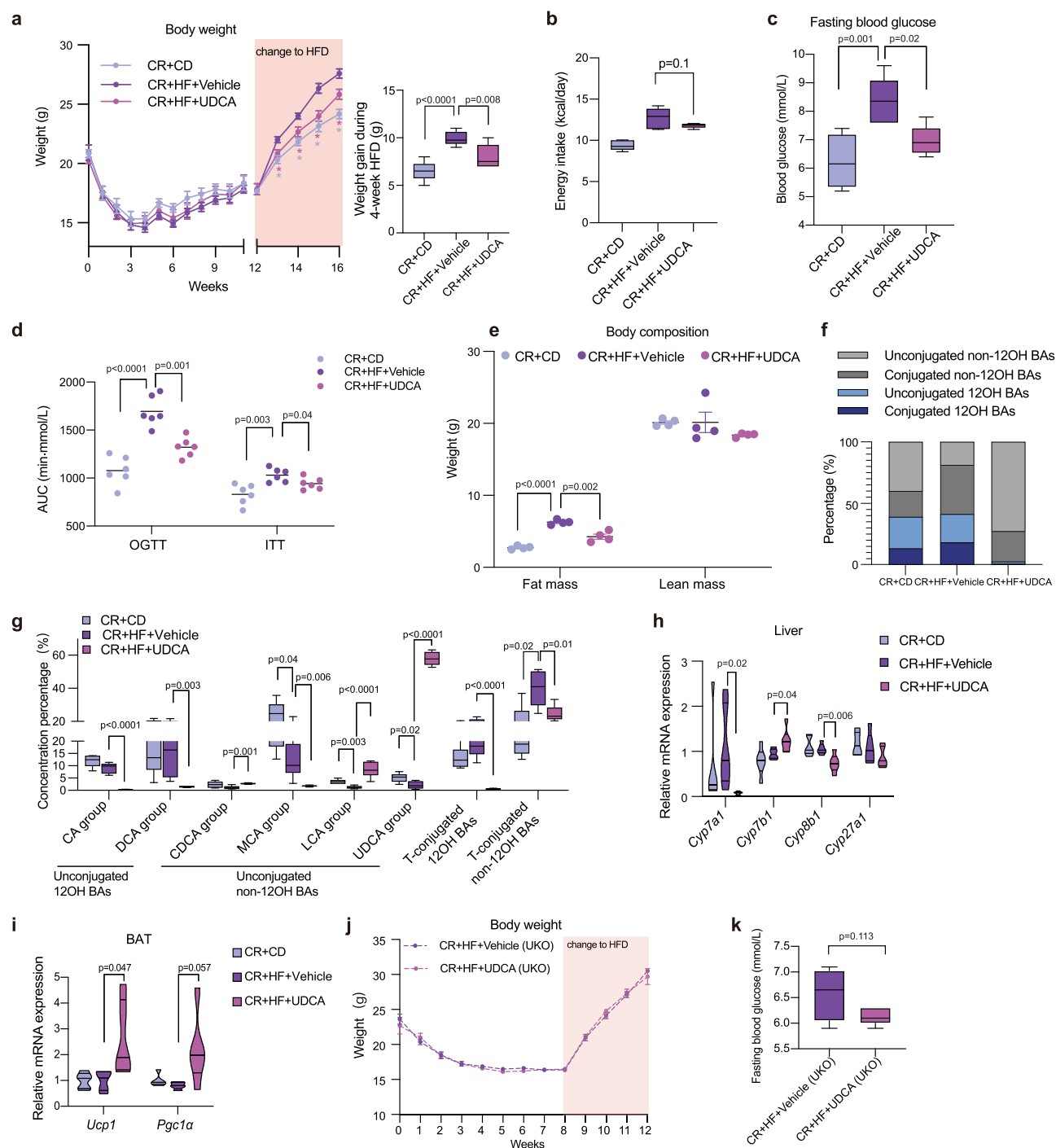

dysbiosis of the gut microbiome. Meanwhile, the enriched valine-leucine-isoleucine biosynthesis pathway indicated the differences in energy source utilization and aging processes during CR.

Calonne et al. found in the rat experiments that the weight regain was due to diminished muscle thermogenesis following CR intervention[34]. In previous studies, the pattern of increased 12OH BAs has also been identified in human patients with diabetes[35,36] and obese or diabetic models in rodents[17,37,38]. Indeed, we also found the same 12OH BAs elevated pattern in our CR diet models, which was consistent with previous studies in CR rodents[39,40]. Moreover, clinical studies have shown that patients with obesity experience an increase in 12OH BAs after 2 weeks of a low-calorie diet[41]. Metabolic syndrome (MetS) patients who have lost weight due to lifestyle also experience serum BA

composition changes to an increased 12OH/non-12OH BAs ratio[42]. Gastric bypass surgery (GBP) can lead to sustained weight loss and significantly improve T2DM. The ratio of 12OH/non-12OH increased significantly after 2 years of GBP[43].

BAs are synthesized to 12OH or non-12OH BAs in the liver via two different pathways. The classic pathway is initiated by CYP7A1-CYP27A1 or CYP7A1-CYP8B1-CYP27A1, which finally generates CDCA or CA, respectively. The alternative pathway is initiated mainly by CYP27A1-CYP7B1 to generate CDCA and the foremost role of this pathway is to generate the regulatory oxysterols that help control cholesterol and lipid homeostasis[44]. After 12-week of CR, CYP7B1 was significantly reduced at both the mRNA and protein level, consistent with decreased alternative pathway activity in the liver. Lower *Cyp7b1*

**Fig. 7 The administration of non-12OH BA attenuated the weight regain after CR diet. a** Body weight (*$p < 0.05$. Differences between the CR + CD group and the CR + HF + Vehicle group are marked in lilac asterisks, differences between the CR + HF + UDCA group and the CR + HF + Vehicle group are marked in purple asterisks.) and the weight gain during the 4-week HFD. **b** Average energy intake during the first week of HFD. **c** Fasting blood glucose level at the end of the experiment. **d** AUC of the OGTT and ITT. **e** Fat mass and lean mass of the mice by NMR miniSpec LF50, $n = 4$ per group. **f** The bar plots show the mean percentage of (non-)12OH and (un)conjugated BAs in the serum among the three groups. **g** The BAs composition profiles in the serum among three groups. $p$-values were adjusted by the FDR's method. **h** Relative mRNA expression of BAs synthesis-related liver enzymes. **i** Relative mRNA expression of $Ucp1$ and $Pgc1\alpha$ in BAT. **j** The body weight of UKO mice. **k** The fasting blood glucose level of UKO mice. $n = 6$ in the CR + CD, CR + HF + Vehicle and CR + HF + UDCA groups. $n = 5$ in the CR + HF + UDCA (UKO) group and $n = 4$ in the CR + HF + Vehicle (UKO) group. All $p$-values in figures were assessed by the two-tailed unpaired T-test (towards CR + HF + Vehicle group) in the GraphPad software. All box and whiskers plots showed the box (from the 25th to 75th percentiles), the median value (in the transverse line), and the whiskers (go down to the smallest value and up to the largest), other data in bar plots were expressed as means ± SEM. Source data are provided as a Source data file. High-fat diet (HFD); ursodeoxycholic acid (UDCA); area under the curve (AUC); oral glucose tolerance test (OGTT); insulin tolerance tests (ITT); 12α-hydroxylated bile acids (12OH BAs); non-12α-hydroxylated bile acids (non-12OH BAs); T-conjugated (taurine-conjugated); cholic acid (CA); deoxycholic acid (DCA); chenodeoxycholic acid (CDCA); muricholic acid (MCA); lithocholic acid (LCA); cytochrome P-450 cholesterol 7α-hydroxylase ($Cyp7a1$); oxysterol 7α-hydroxylase ($Cyp7b1$); sterol-12α-hydroxylase ($Cyp8b1$); cytochrome P-450-27A1 ($Cyp27a1$); brown adipose tissue (BAT); uncoupling protein 1 ($Ucp1$); peroxisome proliferator-activated receptor-γ coactivator-1α ($Pgc1\alpha$).

mRNA expression was also observed in mouse models of insulin resistance and T2DM mice[38,45,46] as well as in patients with obesity and T2DM[47]. Knockout of CYP7B1 resulted in significantly higher lipid content in BAT[48], and elevated plasma and tissue levels of 25- and 27-hydroxycholesterol[49]. CDCA is transformed into UDCA and LCA by *Parabacteroides distasonis* and other intestinal flora. The treatment of the mixture LCA and UDCA was also shown to alleviate obesity and improve glucose and lipid homeostasis in HFD mice[30]. UDCA antagonizes FXR and LCA is a potent activator of the TGR5 signaling pathway[12] which regulates GLP-1 secretion and improves glucose homeostasis[50]. The imbalance of energy intake and energy expenditure can lead to obesity development. LCA and other TGR5 agonists that increase $Ucp1$ expression in BAT and enhance energy expenditure is a potential strategy for preventing obesity. Here, we saw the $Tgr5$ expression decreased in the CR + HF group, which can lead to metabolic changes that increase weight regain. Other potential mechanisms of thermogenic program upregulation may provide additional therapeutic options (e.g., Fgf21, irisin, increased SNS) and require further investigations. It should be noted that due to the increased synthesis of total BA after CR, many 12OH BAs (such as CAs and DCAs) were increased to a larger proportion (Supplementary Fig. 3b). However, the absolute concentrations of non-12OH BAs (such as UDCAs) remained unchanged. We believe that the BA pool is influenced by many factors and that individual BAs impact the BA receptor signaling pathways via their ability to act as either agonists or antagonists. Therefore, the form of BA composition is increasingly used in BA data analysis and presentation[51–55], as in this study, providing more comprehensive results to display integrated biological functions of BAs.

There are several limitations to this study. First, all of the animal models used were male, the sex-based confounder in gut microbial and BAs composition analyses were ignored. Second, we did not detect short-chain fatty acids in the intestinal contents, which could be produced by gut microbes and also influence glycolipid metabolism. Due to equipment limitations, all the experiments were established at the typical laboratory room temperature (22–25 °C), and the phenomenon of the weight regain model at thermoneutrality (29 °C) needs further investigation. Further studies in human dieting and weight rebound are required.

Overall, our study revealed a link between the rebound weight gain after CR and a more efficient energy-harvesting metabolic phenotype involving a unique gut microbiota structure and BA composition. We found a probiotic gut microbiota, *Parabacteroides distasonis*, diminished during CR that could affect the proportion of secondary non-12OH BAs and the metabolism of glucose and lipid during the weight regain. *Parabacteroides distasonis* has the potential as a nutritional supplement to replenish beneficial bacteria that have been lost by people who have lost weight due to CR dieting.

## Methods

**Animal studies.** The animal studies were complied with all relevant ethical regulations and approved by the Institutional Animal Care and Use Committee of Shanghai Jiao Tong University Affiliated Shanghai Sixth People's Hospital, and the animal welfare ethics acceptance number is DWLL2020-0571. C57BL/6J mice (Specific pathogen-free grade, male, 6 weeks old) were purchased from Shanghai Laboratory Animal Center (SLAC, Shanghai, China) and allowed 1-2 weeks of acclimatization. Mice were maintained under a controlled environment (22–25 °C, 12-h light/dark cycle) and were housed in individually ventilated cages under specific pathogen-free conditions with free access to ultrapure water. The chow diet (TP23522, Trophic Animal Feed High-tech Co., Ltd, China) or HFD (D12492, Research Diets, Inc) are used in all mice experiments.

**Animal experiment 1.** A total of 36 mice were randomly divided into two groups with chow diet and HFD for 6 weeks. The diets used in this study are purified ingredient diets and not grain-based diets, differences between groups could be related to the level of fat. At the end of the sixth week, half of the total mice in each group underwent an OGTT and were euthanized later for liver, serum, and cecal content samples collection. The two groups of samples were defined as N-BF and H-BF. The rest 18 mice ($n = 9$ per group) were all fed the chow that provided 50% calories of their baseline need for 4 days, followed by complete fasting for 4 days, and then another 50% calorie restriction for 4 days. In the following 4 weeks, mice were fed the chow diet ad libitum. All of the mice underwent an OGTT at the end of the 4 weeks and were euthanized for liver, serum, and cecal content samples collection. The two groups of samples were defined as N-RF and H-RF.

**Animal experiment 2.** The mice were randomly divided into three groups with different diets for 12 weeks ($n = 8$ per group). The chow diet group (CD group) was fed a chow. The HFD group (HF group) was fed with open-source diet D12492. The CR group was fed with 60% calories of the CD group per day. Body weight and dietary intake were monitored during the experiments.

**Animal experiment 3.** The mice were randomly allocated into three groups and administered either chow, HFD, or CR diet for 12 weeks at which time, all three groups were placed on HFD for an additional 4 weeks ($n = 8$ per group) or 8 weeks for the observation of weight regain ($n = 6$ per group). Body weight and dietary intake were monitored during the experiments. In the 12th and 16th weeks, mice in each group underwent OGTT and ITT. The two tests were conducted at least 4 days apart in order to allow recovery of the mice. Fecal samples were collected within 24 h at the end of the experiments to determine fecal energy excretion using the calorimeter, IKA C5000.

**Animal experiment 4.** The mice were fed chow diet, randomly allocated into three groups ($n = 8$ in C-trans and R-trans group, $n = 5$ in H-trans group) and administered antibiotic mixtures (including vancomycin: 100 mg/kg/day, neomycin: 200 mg/kg/day, metronidazole: 200 mg/kg/day, ampicillin: 200 mg/kg/day) for 3 weeks to model the antibiotic-treated-germ-free animal models. Feces of the donors were collected and dispersed in sterile phosphate buffer saline (PBS) buffer,

using the supernatant for transplantation. Then all three groups were placed on HFD for an additional 4 weeks and were orally gavaged with fecal suspension from mice in HF + HF (for H-trans), CD + HF (for C-trans), and CR + HF (for R-trans) groups. The body weight was recorded once a week.

**Animal experiment 5**. The mice were randomly allocated into three groups and administered CR diet for 12 weeks at which time, all three groups were placed on HFD along with either supplement of *Parabacteroides distasonis*, PBS, or heat-killed *Parabacteroides distasonis* for an additional 4 weeks (*n* = 5 per group). The dosages for the groups of mice were (1) oral administration of 200 μL *Parabacteroides distasonis* (5 × 10⁹ cfu/mL in sterilized PBS) per mouse daily (CR + HF + PD group), (2) oral administration of 200 μL sterilized PBS per mouse daily (CR + HF + Vehicle group), (3) oral administration of 200 μL heat-killed *Parabacteroides distasonis* (5 × 10⁹ cfu/mL in sterilized PBS) (CR + HF + HKPD group) per mouse daily. Body weight and dietary intake were monitored during the experiments. In the 12th and 16th weeks, mice in each group underwent OGTT and ITT. The two tests were conducted at least 4 days apart in order to allow recovery of the mice. Fecal samples were collected within 24 h at the end of the experiments to determine fecal energy excretion using the calorimeter, IKA C5000.

**Animal experiment 6**. Eighteen mice were randomly divided into three groups and administered CR diet for 12 weeks, then with different diets feeding for 4 weeks: (1) chow diet group (CR + CD), (2) HFD with vehicle (CR + HF + Vehicle) group, and (3) HFD with 0.5% (w/w) UDCA (CR + HF + UDCA) group. *n* = 6 per group. Body weight and food intake were recorded once a week during the experiments. The OGTT and ITT test were performed at the end of the experiment. The two tests were conducted at least 4 days apart in order to allow recovery of the mice. Fecal samples were collected within 24 h at the end of the experiments to determine fecal energy excretion using the calorimeter, IKA C5000.

**Animal experiment 7**. Fourteen 6–8 weeks female UCP1-knockout (UKO) mice were randomly divided into three groups and administered a CR diet for 8 weeks, followed by HFD feeding for 4 weeks: (1) HFD with oral administration of 200 μL sterilized PBS per mouse daily (CR + HF + Vehicle (UKO)), *n* = 4, (2) HFD with oral administration of 200 μL *Parabacteroides distasonis* (5 × 10⁹ cfu/mL in sterilized PBS) per mouse daily (CR + HF + PD (UKO)), *n* = 5, and (3) HFD with 0.5% (w/w) UDCA per mouse daily (CR + HF + UDCA (UKO)), *n* = 5. Body weight and food intake were recorded once a week during the experiments.

**Animal experiment 8**. Ten mice were randomly divided into 2 groups and administered a CR diet for 8 weeks, followed by HFD feeding for 4 weeks. The mice were intraperitoneally injected with GLP-1 receptor agonist, exendin-4 (exenatide, 24 nmol/kg, Cas: 141758-74-9, MedChemExpress), or PBS control once daily. *n* = 5 per group. Body weight and food intake were recorded once a week during the experiments.

After all experiments, the mice were fasted overnight and then euthanized for harvesting blood and other tissues. Blood was centrifuged at 2200 × *g* for 10 min for serum isolation. Liver samples, intestinal tissues and contents, white and brown fat tissues were collected immediately after euthanasia. All samples were stored at −80 °C until used for analysis.

**Metabolic cages**. At week 16th, four mice per group were randomly selected and individually placed in metabolic cages (PROMETHION BX1, Sable Systems International, USA). The mice were acclimated to the system for 48 h before formal testing. The studies were maintained at 22–25 °C with a 12h light/dark cycle and the mice could get free access to food and water.

**Sequencing of the whole microbial genome**. Five samples of cecal feces from each of the three groups (CD, HF, CR) were randomly selected for whole microbial genome sequencing analysis. Microbial genome DNA was extracted from these samples by DNeasy PowerSoil Pro Kit (Cat: 47016, QIAGEN, Inc., Netherlands) using the manufacturer's protocol. Successful DNA isolation was confirmed by using a NanoDrop ND-1000 spectrophotometer (Thermo Fisher Scientific, Waltham, MA, USA) and agarose gel electrophoresis. Metagenomic high-throughput sequencing was performed using the Illumina platform (Illumina, USA) with the PE150 whole-genome shotgun strategy at Personal Biotechnology Co., Ltd. (Shanghai, China). Raw sequencing was saved in FASTQ format and processed to obtain quality-filtered reads. The sequencing adapters were removed from sequencing reads using Cutadapt (v1.2.1), then low-quality reads were trimmed by using a sliding-window algorithm. To remove host contamination, reads were aligned to the host genome using BWA (http://bio-bwa.sourceforge.net/). The quality-filtered reads were de novo assembled to construct the metagenome for each sample by IDBA-UD. All coding regions of metagenomic scaffolds longer than 200 bp were predicted by MetaGeneMark. The functional profiles of the non-redundant genes were obtained by annotation against the KEGG and Evolutionary Genealogy of Genes: Non-supervised Orthologous Groups (EggNOG) databases. To obtain the relative abundance distribution of each sample at each classification level (phylum, class, order, family, genus, species), we used BLASTN to align the

Scaffolds/Scaftigs sequences with the bacteria in the NCBI database (E-value <0.001). To preserve the biological significance, we adopted the "lowest common ancestor" algorithm in the MEtaGenome Analyzer (MEGAN) software.

**16S rRNA gene sequencing**. Seven cecal feces samples per group from N-BF, H-BF, N-RF, and H-RF groups were randomly selected for microbiota 16S rRNA analysis. All cecal feces samples from CR + HF + PD, CR + HF + Vehicle, CR + HF + HKPD groups were selected for microbiota 16S rRNA analysis. Microbial genome DNA was extracted from these samples by using a QiaAmp DNA stool Mini Kit (Cat: 51604, QIAGEN, Inc., Netherlands) following the manufacturer's protocol. Successful DNA isolation was confirmed by using agarose gel electrophoresis. The V4-V5 hypervariable regions of 16S rRNA were PCR amplified from the microbial genomic DNA harvested from the cecal samples and were used for the remainder of the study. PCR amplification of the V4-V5 region of bacterial 16S rRNA genes was performed using the forward primer (5′-GTGCCAGCMGC CGCGGTAA-3′) and the reverse primer (5′-CCGTCAATTCMTTTRAGTTT-3′). After sequencing, bacterial OTUs were counted for each sample to express the richness of bacterial species with an identity cutoff of 97%. Bacterial OTUs were generated by the uclust method in the QIIME (http://qiime.org/scripts/pick_otus.html). Taxon-dependent analysis was conducted by using the Greengene database.

**BAs analysis**. BAs were quantified using in-house established methods. For serum samples pretreatment, an aliquot of 40 μL serum sample was mixed with 240 μL acetonitrile–methanol (8:2, v/v, with d4-CA, d4-UDCA, d4-LCA, d4-GCA, d4-GCDCA, d4-GDCA each 50 nM). The mixture was allowed to stand at −20 °C for 30 min, then centrifuged at 19,090 × *g* at 4 °C for 15 min. A total of 240 μL supernatant was transferred to another tube and then vacuum-dried. An aliquot of 40 μL volume of acetonitrile–methanol (9:1, v/v) containing 0.01% formic acid and then 60 μL of ddH₂O was added. The sample was vortexed and then centrifuged at 19,090 × *g*, at 4 °C for 15 min. The supernatant was used for UPLC/TQ-MS analysis. For liver and cecal contents samples, 10 mg of samples were weighed and homogenized with 200 μL of a mixture of acetonitrile–methanol (1:1, v/v, with d4-CA, d4-UDCA, d4-LCA, d4-GCA, d4-GCDCA, d4-GDCA each 50 nM) for 5 min. The supernatant was transferred after centrifugation at 19,090 × *g* for 15 min. The residue was reconstituted with 200 μL of a mixture of acetonitrile–methanol (8:2, v/v, with d4-CA, d4-UDCA, d4-LCA, d4-GCA, d4-GCDCA, d4-GDCA each 50 nM) for 5 min. After centrifugation at 19,090 × *g* for 15 min, the supernatant was transferred to the same tube with the first-step supernatant. Each combination was centrifuged at 19,090 × *g* for 10 min and the supernatant was used for UPLC/TQ-MS analysis. All of the samples were run in a randomized order to minimize systematic analytical errors. Samples were selected randomly from each group and were pooled as quality control samples. The peak annotation and quantification were performed by MassLynx v4.1 and TargetLynx v4.1 (Waters Corp., Milford, MA, USA). 12OH BAs detected in our study included CA group (CA, 3-ketoCA), DCA group (DCA, 7-ketoDCA, 23-norDCA), and their taurine-conjugated and glycine-conjugated forms; all other BAs detected were non-12OH BAs that included muricholic acids group (αMCA, βMCA, ωMCA), UDCA group (UDCA, βUDCA), LCA group and their taurine-conjugated and glycine-conjugated forms.

**Microbial metabolites analysis**. The quantitative analysis was using UPLC/TQ-MS (LC 1290, MS 6460, Agilent Technologies, USA) according to a protocol we previously established[56]. For serum samples pretreatment, 5 μL of each serum sample was mixed with 25 μL 160 mM 3-nitrophenylhydrazine-hydrochloride and 25 μL of 40 mM N-Ethyl-N′-(3-dimethylaminopropyl) carbodiimide hydrochloride (EDC·HCl) solution. The mixture was vertexing at 30 °C for an hour, and then the reaction solution was stored under −20 °C for 20 min and centrifugated at 4 °C for 15 min, 19,090 × *g*. The calibration curve was prepared in the same way as described above. The peak annotation and quantification were performed by Agilent MassHunter (SCN 712, Agilent Technologies, USA). A total of 123 microbial metabolites were quantitatively measured, including fatty acids, amino acids, organic acids, carbohydrates, indoles, carnitines, and BAs.

**Microbial strain**. *Parabacteroides distasonis* was purchased from the Guangdong Microbial Culture Collection Center (GDMCC 1.1564) and was identified by comparing the 16S rRNA gene sequences with the NCBI reference database, it showed 100% similarity to the sequence of *Parabacteroides distasonis* ATCC 8503 (NCBI Reference Sequence: NC_009615.1). The sequences are shown in Supplementary Table 1. The strain was cultured using tryptic soy agar/broth (Cat: BD 236950/BD 211825, BD Difco™, USA) with 5% sterile defibrinated sheep blood (Cat: FR-10013, Kangrun Biological Technology Co., Ltd., Shanghai, China) under anaerobic conditions (80%N₂, 10%CO₂, 10%H₂). The bacteria were enriched by centrifuging at 1400 × *g* for 10 min at room temperature and suspended in oxygen-free sterilized PBS with a final density of 5 × 10⁹ cfu/mL.

**RNA, DNA isolation, and quantitative real-time PCR**. The distal ileum and liver tissues were homogenized, and total RNA was isolated using TRIzol Reagent (Cat: 15596026, Invitrogen, CA, USA). The DNA of cecal contents for PCR was isolated using QIAamp Fast DNA Stool Mini Kits (Cat: 51604, QIAGEN, Inc., Netherlands)

by following the manufacturer's protocol. The total RNA and dsDNA concentration were measured using a NanoDrop™ One spectrophotometer (Thermo Scientific, MA, USA). 500 ng total RNA from each ileal/liver sample were reverse transcribed to form the cDNA templates using the Prime Script RT Reagent Kit (Cat: RR037A, TAKARA, Kusatsu, Japan). The qPCR primers were designed and synthesized (Sangon Biotech, Shanghai, China) and the sequences are shown in Supplementary Table 2. ChamQ Universal SYBR qPCR Master Mix (Cat: Q711, Vazyme Biotech Co., Ltd., China) was used for quantitative real-time PCR by Applied Biosystems QuantStudio 7 Flex System (Thermo Scientific, MA, USA). The relative expression levels of the target genes were normalized to *Gapdh* or *16s* and analyzed using the $\Delta\Delta$CT analysis method.

**Western blot analysis**. The liver and BAT tissues were lysed with RIPA buffer (Cat: P0013B, Beyotime Technology, Shanghai, China) containing 1 mM PMSF (Cat: ST506, Beyotime Technology, Shanghai, China). The protein concentrations were quantified using the BCA Protein Assay Kit (Cat: 23225, Pierce, Rockford, IL, USA). The denatured proteins were resolved by 10% SDS-page gels and transferred to Immobilon-P Transfer Membranes (Millipore Corporation, Tullagreen, IRL). The membranes were blocked at room temperature with 5% nonfat milk and the membranes were incubated with antibodies against UCP1(1:1000, Cat: #14670, Cell Signaling Technology, Beverly, MA), CYP7B1 (1:1000, Cat: ab138497, Abcam, Cambridge, United Kingdom), and β-Actin (1:1000, Cat: #4970, Cell Signaling Technology, Beverly, MA) at 4 °C overnight, then incubated with horseradish peroxidase-conjugated secondary antibodies (Anti-rabbit IgG, Cat: #7074, Cell Signaling Technology, Beverly, MA). The bands were visualized using a Tanon™ High-Sig ECL Western Blotting Substrate (Cat: sb-wb011, Tanon Science & Technology Co., Shanghai, China) in the BIO-RAD ChemiDoc MP imaging system (Bio-Rad, CA). The gray values of the bands were calculated using ImageJ software (version 1.53) and were normalized to β-Actin.

**Histopathological examination**. Samples of the liver and WAT were resected and fixed in 4% paraformaldehyde and embedded in paraffin according to standard procedures. The samples embedded in paraffin were sectioned and further stained with H&E staining. Images were acquired using an inverted fluorescence microscope ECLIPSE TS2R (Nikon, Tokyo, Japan).

**GLP-1 active quantification**. The mice were fasted overnight, GLP-1 inhibitor sitagliptin was gavaged into mice (3 mg/kg). One hour later, a liquid feed of enteral nutrition emulsion (1 g in 3 mL ddH$_2$O) was gavaged into the mice (100 μL/10 g). After 15 min, we collected blood from the retro-orbital. Blood was centrifuged at $2200 \times g$ for 10 min for serum isolation. The serum GLP-1 levels were quantified using ELISA kits (Cat: EGLP-35K, Millipore, MA, USA) according to the manufacturers' instructions. The optical density at ~425 nm was measured using a SpextraMax i3x Multi-mode Microplate Reader (Molecular Devices, USA). A four-parameter logistic curve fit was selected to generate the standard curve and calculate the concentration of each sample.

**Statistical analysis**. All the bar plots in this study were generated using GraphPad Prism 8.4.0 (GraphPad Software, San Diego, USA). The differential analysis was performed using the two-tailed unpaired T-test and Kruskal–Wallis test in the GraphPad Prism 8.4.0. IP4M 2.0 (http://ip4m.cn) was also used in the data processing. Correlations between BAs and microbiome abundances were performed using GRaMM strategy in R package "gramm4R" (R version 3.6.3)[22]. PLS-DA and hierarchical clustering were conducted in R package "mixOmics" (R version 3.6.3). PCoA was generated in R package "vegan" (R version 3.6.3). Differences between experimental groups were considered significant at $p < 0.05$.

**Reporting summary**. Further information on research design is available in the Nature Research Reporting Summary linked to this article.

## Data availability

The 16S rRNA gene sequences and metagenomic sequences were provided and available at National Center for Biotechnology Information Sequence Read Archive (SRP) database with accession code PRJNA813171 and PRJNA813023. The metabolomics data were deposited and available at MetaboLights repository with accession code MTBLS4431. Source data are provided with this paper.

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

## Acknowledgements

This work was supported by the National Key R&D Program of China 2021YFA1301300 (W.J.) and the National Natural Science Foundation of China 82122012 (X.Z.), 81974073 (W.J.), and 31972935 (T.C.).

## Author contributions

W.J. conceptualized and designed the study. M.L., X.Z., and W.J. coordinated the experimental planning and execution. M.L., M.Z., J.K., J.W., and D.L. were responsible for mouse experiments. M.L. and A.Z. were responsible for sample preparation and BA analysis. M.L. performed the data preprocessing and statistical analysis. M.L., X.Z., and W.J. drafted the manuscript and produced the figures. S.W., J.K., Y.L., J.W., C.R., A.Z., and W.J. critically revised the manuscript. M.W., Y.T., Z.R., T.C., X.M., C.H., C.S., W.P.J., and P.L. provided valuable suggestions.

## Competing interests

The authors declare no competing interests.
