## [Peer Review File · Nature Communications]

REVIEWER COMMENTS

Reviewer #1 (Remarks to the Author):

Li et al. have explored a mouse model in which a period of fasting is followed a period of high-fat feeding, which results in weight regain and impaired glucose tolerance. The authors then explore the post-fasting microbiome and bile acids as mechanistic candidates mediating this phenotype. They propose a pathway by which caloric restriction leads to the reduction of Parabacteroides distasonis in the gut, decreased levels of non-12OH bile acids in the serum, reduced serum GLP-1 and diminished expression of Ucp1 in brown adipose tissue.

The novelty is somewhat compromised by a study from the Elinav group several years ago which linked post-dieting weight regain to the microbiome, flavonoid metabolites, and Ucp1. However, with its focus on bile acids, this study is nonetheless a very important contribution to the field. The major concern with this manuscript is the lack of causality in certain instances where causality is claimed. Below are several aspects that should be improved.

Major comments:

1. Regarding the experimental setup, the authors posit that weight gain in the CR+HF group is enhanced compared to the Control+H. However, they don't follow the weight curve for long enough to evaluate whether they simply "catch up" and regain the formerly lost weight, or whether they surpass the control group due to the microbiome-mediated effects that the authors have uncovered.
2. Regarding causality, the authors make the claim that "treatment with Parabacteroides distasonis or non-12OH BAs ameliorated weight regain in CR mice via increased thermogenesis". This is implied by Ucp1 expression but not directly proven. Several pieces of experimental evidence need to be provided to bolster this statement:
 - a) Ucp1 should be examined on the protein level, since mRNA levels do not always correlate with protein expression for this molecule.
 - b) Energy expenditure should be measured as a direct functional readout of enhanced Ucp1 expression.
 - c) Performing the experimental paradigm on Ucp1 KO or BATectomized mice may be beyond the scope of this study, but would be the gold standard for demonstrating causality.
3. Causality may be easier to establish in cases where there are pharmacological tools to interfere with the proposed pathway. This can be done, for instance, for GLP-1. If GLP-1 is indeed, as the authors suggest, a causative mediator of the microbiome/bile acid effect, then a GLP-1 receptor agonist should rescue the metabolic derangements in CR+HF mice.

Minor comments:

1. The abstract is a bit hard to follow since the weight regain model is not introduced, and thus "the weight-rebound mice" come out of nowhere. It may be better to mention the experimental model early on for more clarity.

2. In Figure 6J, the gene name should be Gcg rather than “glucagon”.

3. The authors may want to modify their statement that “The altered gut microbiota resulting from CR was similar to those exposed to HFD.” This applies to the specific case of *Parabacteroides distasonis* and several other species, but Figure 2A shows that the communities are still strongly distinct.

Reviewer #2 (Remarks to the Author):

In this paper, the authors have examined changes in the microbiome, bile acids, and gene expression in models of weight regain. Generally speaking, HF feeding and calorie restriction ((CR) led to similar changes in the microbiome and BA metabolite profiles in the gut and serum. A decline in *Parabacteroides distasonis* (PD) was common for both, and transplant and supplement studies with PD or the non-12OH bile acids attenuated weight regain. Overall, the authors present evidence of a role of the gut microflora in mediating some aspects of the biological drive to regain weight, and honed in on a mechanism that engages PD, bile acids, and their effect on gene expression in peripheral tissues. I have a number of questions and comments for the authors to consider.

Major concerns

- The authors also need to justify the “fasting” paradigm of 4 days of 50% CR, followed by 4 days of 100% CR, followed by 50% CR. What is this experimental paradigm supposed to model? This undoubtedly would have sent the mice into a starvation period that would have resulted in a significant amount of lean mass loss and possibly torpor. The premise for pursuing and including this extreme starvation intervention and protein malnourishment is unclear, and body composition measures would be important for interpreting the data.
- The authors suggest in several places (abstract, results, discussion) that they have measures of energy expenditure. While the data presented may suggest there may be coincident changes in energy expenditure to explain their observations, no such data is presented. Ideally, the authors should present this data to support this hypothesis; alternatively, they need to substantively temper the language throughout so that presentation better reflects what was measured.
- In general, more comprehensive measures of energy balance and body composition are need to support the authors claims about thermogenesis. Most specifically, a more refined temporal presentation of EI in all of the studies would strengthen the claim that appetite regulation was not involved.
- The studies appear to be performed in relatively young, growing mice. Given they are still in the phase of laying down lean mass, it would seem these studies would be more relevant to the “Catch-up” growth phenomenon put forth by Dulloo et al, rather that weight regain phenomenon commonly seen with dieting in humans. Stunting lean mass growth would change the metabolic requirements of the animals. Figures S1D and S1E are difficult to interpret without body composition data.

Additional comments and concerns

- Fig S1B. The authors should justify the diets. They appear to be semi-purified diets which vary in a

number of aspects that go beyond the macronutrient composition, as well as the relevance of a 60% fat diet.

- Ln 30. Thermogenesis was not measured.
- Ln 55. The acronym OUT needs to be defined early in the manuscript.
- Ln 95-98. More increment body weight curves would be more informative.
- Fig 1A and others. The text is exceptionally small in the workflow figures. Even zooming to 200%, I had trouble reading the text.
- Ln 184-185. The interpretation is overstated- the coincident observations would suggest causality, but they do not provide definitive evidence that the changes were the result of lower CYP7B1 expression.
- Fig 1D. The authors need to clarify how EI was measured and calculated.
- Ln 203-204. The authors are claiming that the fecal transplant is reducing energy expenditure, but no measures of intake or expenditure are presented to show what might be happening with energy balance.
- Fig 1G. Need to clarify the units of the relative gain (presumably g/day).
- Ln 218-219. The header should not imply that energy expenditure was changed since it was not measured.
- Ln 247-248. How was energy intake measured and calculated for this experiment? Much of the difference in body weight occurred during the first few days- so the informative measures should come from this early period. Averaging intake over the 30 day follow-up would miss an effect on intake during this early window.
- Fig 7B. More incremental measures of EI would be informative, particularly during the window when differences in weight gain occurred. If the data are skewed by measurements toward the end of the study, the differences are likely just a reflection of that body mass differences (metabolic requirements. Again, body composition data would strengthen the studies and their interpretation in support of the fat pad weights presented.
- Fig S1F. Morphology from an H&E stain image was used to suggest steatosis—it would be more convincing to have a lipid stain to support this claim.
- Ln 548-549. Please clarify – you collected blood from the eyeballs? Perhaps you mean from a retro-orbital bleed?
- Ln 395. The authors should consider in the limitations how studying these mice well below thermoneutral conditions may have affected these studies of energy balance, particularly as they are making claims with regard to thermogenesis and energy expenditure.
- Ln 572. It is unclear throughout the paper why the authors jump between parametric to nonparametric tests. Broadly speaking, the whisker box-plots with individual datapoints would be preferred throughout the paper, rather than the bar graphs. The authors should clarify which statistical approach was used for each experiment. In a number of cases, consider presenting absolute values rather than fold change.
- Figures S6A and s7A appear to be the same with an alternate color scheme. Fig S7A should describe the UCDA experiments.

Reviewer #3 (Remarks to the Author):

The current submission described a deep and meaningful investigation on the bodyweight rebound after CR treatment. It is an interesting work providing new insights into the mechanism underlying the bodyweight regain after fasting or CR. The experiments design and analysis methods are solid to support the final conclusion. Parabacterial distasis is the key mediator for the control of body weight. Loss of this gut commensal bacteria after CR is demonstrated to be the causes for bodyweight regaining. And also, alteration of non12-OH bile due to the change of gut microbiome, especially the decrease of P. distasis, is confirmed as the gut metabolites mediating the recovery effects. This work is well organized and clearly written.

To my judgement, it can be accepted in its current version.

REVIEWER COMMENTS

Reviewer #1 (Remarks to the Author):

Li et al. have explored a mouse model in which a period of fasting is followed a period of high-fat feeding, which results in weight regain and impaired glucose tolerance. The authors then explore the post-fasting microbiome and bile acids as mechanistic candidates mediating this phenotype. They propose a pathway by which caloric restriction leads to the reduction of *Parabacteroides distasonis* in the gut, decreased levels of non-12OH bile acids in the serum, reduced serum GLP-1 and diminished expression of Ucp1 in brown adipose tissue.

The novelty is somewhat compromised by a study from the Elinav group several years ago which linked post-dieting weight regain to the microbiome, flavonoid metabolites, and Ucp1. However, with its focus on bile acids, this study is nonetheless a very important contribution to the field. The major concern with this manuscript is the lack of causality in certain instances where causality is claimed. Below are several aspects that should be improved.

Thanks for your positive comments. We are aware of the very nice work done by the Elinav group (Nature, 540, 7634: 544-551, 2016) demonstrating the microbiome contribution to diminished post-dieting flavonoid levels and reduced energy expenditure. We have revised the manuscript carefully with additional experimental results to strengthen the causal role of microbiota – bile acid interactions in post dieting weight gain, according to your valuable suggestions.

Major comments:

1. Regarding the experimental setup, the authors posit that weight gain in the CR+HF group is enhanced compared to the Control+H. However, they don't follow the weight curve for long enough to evaluate whether they simply "catch up" and regain the formerly lost weight, or whether they surpass the control group due to the microbiome-mediated effects that the authors have uncovered.

Answer:

Thank you for your comments. Actually, we have observed the body weights for longer time, and found that after 6-week HFD treatment (data shown as the 18th week in the figure), there was no significant difference between the CR+HF group and CD+HF group (Control group). We have added this part of the results into the manuscript (Ln133-134).

Figure (revised Figure S1H). Body weights during experiment and relative weight gain rate during the period of change to HF diets. n=6 per group. Data are expressed as means±SEM. Differences were assessed by the Student's T-test, *p<0.05; **p<0.01; ***p<0.001; #p<0.0001; n.s: no significance. CD: chow diet; HF: high-fat diet; CR: calorie restriction diet.

2. Regarding causality, the authors make the claim that "treatment with Parabacteroides distasonis or non-12OH BAs ameliorated weight regain in CR mice via increased thermogenesis". This is implied by Ucp1 expression but not directly proven. Several pieces of experimental evidence need to be provided to bolster this statement:

- Ucp1 should be examined on the protein level, since mRNA levels do not always correlate with protein expression for this molecule.
- Energy expenditure should be measured as a direct functional readout of enhanced Ucp1 expression.
- Performing the experimental paradigm on Ucp1 KO or BATectomized mice may be beyond the scope of this study, but would be the gold standard for demonstrating causality.

Answer:

Thanks for your advice. Accordingly, we have examined the protein level of UCP1, and measured the energy expenditure in the experiments with *P. distasonis* or UDCA intervention. The Ucp1 KO mice have also been applied for demonstrating the causality. Moreover, as the energy expenditure is quite important data to support the observation of thermogenesis, the data have also been added in the experiment with HF, CD, CR, and 4-week HF intervention (revised Fig 1I). The data are shown in the following answers and revised manuscript.

- The protein expression of UCP1 was detected, treatments with *P. distasonis* or UDCA elevated the UCP1 expression in the CR+HF mice (revised Figure 6N, 7K).

Figure (revised Figure 6N). UCP1 protein expression in BAT. Differences were assessed by the Student's T-test, ** $p < 0.01$; *** $p < 0.001$.

Figure (revised Figure 7K). UCP1 protein expression in BAT. Differences were assessed by the Student's T-test, * $p < 0.05$.

b) Energy expenditures were assessed in three experiments, the results were added into the manuscript (revised Figure 6C, 7C, and 1I). Treatments with *P. distasonis* or UDCA elevated energy expenditure (revised Figure 6C, 7C).

Figure (revised Figure 6C). Energy expenditures in the *P. distasonis* treatment experiments. n=4 per group. Differences were assessed by the two-way analysis of variance (ANOVA), *p<0.05 and **p<0.01.

Figure (revised Figure 7C). Energy expenditures in the UDCA treatment experiments. n=4 per group. Differences were assessed by the two-way analysis of variance (ANOVA), *p<0.05.

Figure (revised Figure 1I). Energy expenditures in the diets changing experiments. n=4 per group. Differences were assessed by the two-way analysis of variance (ANOVA), *p<0.05.

c) We performed the experiments of Ucp1 KO (UKO) mice and found no significant weight regain changes at the treatment with *P. distasonis* or UDCA comparing with the CR+HF+Vehicle (UKO) group. We added these results to the revised manuscript (revised Figure 6O, 7L).

Figure (revised Figure 6O (left) ,7L (right)). The body weights of CR+HF (UKO) mice. n=5 in CR+HF+PD (UKO) and CR+HF+UDCA(UKO) group, n=4 in CR+HF+Vehicle (UKO) group. Differences were assessed by the Student's T-test, n.s: no significance.

3. Causality may be easier to establish in cases where there are pharmacological tools to interfere with the proposed pathway. This can be done, for instance, for GLP-1. If GLP-1 is indeed, as the authors suggest, a causative mediator of the microbiome/bile acid effect, then a GLP-1 receptor agonist should rescue the metabolic derangements in CR+HF mice.

Answer:

Thank you for your comments. We have performed the experiments of GLP-1 receptor agonist (exendin-4) to rescue the metabolic derangements in CR+HF mice. The results showed that the weight regain was significantly lower during the following HF diets period after GLP-1 receptor agonist treatment. We have added this result in the manuscript (revised Figure S5G).

Figure (revised Figure S5G). Body weights. n=5 per group. Differences were assessed by the Student's T-test, **p<0.01; #p<0.0001.

Minor comments:

1. The abstract is a bit hard to follow since the weight regain model is not introduced, and thus “the weight-rebound mice” come out of nowhere. It may be better to mention the experimental model early on for more clarity.

Answer: Thanks for your suggestions, and the abstract has been revised accordingly: “...after resuming food consumption” (Ln22).

2. In Figure 6J, the gene name should be Gcg rather than “glucagon”.

Answer: Thank you for your suggestion, and we have changed the gene name to “Gcg” in the figure (revised Figure 6).

3. The authors may want to modify their statement that “The altered gut microbiota resulting from CR was similar to those exposed to HFD.” This applies to the specific case of Parabacteroides distasonis and several other species, but Figure 2A shows that the communities are still strongly distinct.

Answer: Thank you for your suggestion, and we have changed the description to: “Several altered gut microbiota resulting from CR was similar to those exposed to HFD” (Ln26-27).

Reviewer #2 (Remarks to the Author):

In this paper, the authors have examined changes in the microbiome, bile acids, and gene expression in models of weight regain. Generally speaking, HF feeding and calorie restriction ((CR) led to similar changes in the microbiome and BA metabolite profiles in the gut and serum. A decline in *Parabacteroides distasonis* (PD) was common for both, and transplant and supplement studies with PD or the non-12OH bile acids attenuated weight regain. Overall, the authors present evidence of a role of the gut microflora in mediating some aspects of the biological drive to regain weight, and honed in on a mechanism that engages PD, bile acids, and their effect on gene expression in peripheral tissues. I have a number of questions and comments for the authors to consider.

Major concerns

- The authors also need to justify the “fasting” paradigm of 4 days of 50% CR, followed by 4 days of 100% CR, followed by 50% CR. What is this experimental paradigm supposed to model? This undoubtedly would have sent the mice into a starvation period that would have resulted in a significant amount of lean mass loss and possibly torpor. The premise for pursuing and including this extreme starvation intervention and protein malnourishment is unclear, and body composition measures would be important for interpreting the data.

Answer:

Thank you for your comments and we have explained the “fasting” paradigm in more detail accordingly. Ln 90-96: “Fasting is a lifestyle without eating any food for a period, for the beneficial effects, such as reduced body weight, delayed aging, and improved health^{18,19}. Meanwhile, there have been some undesirable outcomes such as weight rebound or the development of food intolerance and inflammation^{20,21}. Therefore, we designed this “fasting” paradigm in mice to mimic stepwise initiation (50% CR for 4 days), the final status of fasting (4 days of 100% CR), and gradual recovery of the food (50% CR for 4 days) by lean or obese individuals.”

We detected the body composition before and after fasting in two kinds of diets (Figure shown below), and we found that fat mass significantly reduced after fasting. The changes of lean mass were not significant, most of the weight loss caused by fasting is the reduction of fat mass.

Figure (revised Figure 1F). Fat mass and lean mass of the mice in the fasting experiment by NMR miniSpec LF50. n=5 per group. Differences were assessed by the Student's T-test, * $p < 0.05$; # $p < 0.0001$, n.s: no significance.

•The authors suggest in several places (abstract, results, discussion) that they have measures of energy expenditure. While the data presented may suggest there may be coincident changes in energy expenditure to explain their observations, no such data is presented. Ideally, the authors should present this data to support this hypothesis; alternatively, they need to substantively temper the language throughout so that presentation better reflects what was measured.

Answer:

Thank you for your comments. We have added several experiments accordingly and added the results in the revised manuscript. Energy expenditure was measured in three experiments (revised Figure 1I, 6C, 7C). We found that CR+HF significantly reduced the energy expenditure (revised Figure 1I) and the *P. distasonis* and UDCA treatment enhanced the energy expenditure in mice (revised Figure 6C, 7C).

Figure (revised Figure 1I). Energy expenditures in the diets changing experiments. n=4 per group. Differences were assessed by the two-way analysis of variance (ANOVA), * $p < 0.05$. CD: chow diet; HF: high-fat diet; CR: calorie restriction diet.

Figure (revised Figure 6C). Energy expenditures in the *P. distasonis* treatment experiments. n=4 per group. Differences were assessed by the two-way analysis of variance (ANOVA), *p<0.05 and **p<0.01.

Figure (revised Figure 7C). Energy expenditures in the UDCA treatment experiments. n=4 per group. Differences were assessed by the two-way analysis of variance (ANOVA), *p<0.05.

•In general, more comprehensive measures of energy balance and body composition are need to support the authors claims about thermogenesis. Most specifically, a more refined temporal presentation of EI in all of the studies would strengthen the claim that appetite regulation was not involved.

Answer:

Thanks for your advice. Accordingly, the energy expenditure, energy intake, and the fecal energy excretion have been detected in the revised experiments. The results showed that during the “weight regain” period, the mice in CR+HF group had the lowest energy intake (Figure shown below), and the most relative weight regain (revised Figure 1G). Fecal energy excretion showed no significant differences among three groups (Figure shown below).

Figure (left: revised Figure S1D, right: revised Figure S1E). Left: Average energy intake per mouse per day during the weight regain period. Right: Fecal energy excretion among the three groups. n=6 per group, n=4 in fecal energy excretion measurements. Differences were assessed by the Student's T-test, *p<0.05; **p<0.01; #p<0.0001; n.s: no significance.

We have also measured the energy intake during the first week of the weight regain period in the experiment with *P. distasonis* or UDCA treatment. The energy intake and fecal energy excretion showed no significant difference after the supplements of PD or UDCA in CR+HF mice (Figures shown below).

Figure (left: revised Figure 6B, right: revised Figure 7B). The average energy intake during the first week of the weight regain period in the experiments of PD intervention (left), and the UDCA intervention (right). n=6 per group. Differences were assessed by the Student's T-test, **p<0.01, n.s: no significance.

Figure (left: revised Figure S5, right: revised Figure S6). Fecal energy excretion in the experiments of PD intervention (left), and the UDCA intervention (right). n=4 per group. Differences were assessed by the Student's T-test, n.s: no significance.

We have also tested the body composition before and after the period of weight regain (Figure shown below). The data showed that the fat mass was significantly lower in CR diet before weight regain period. After weight regain period, the fat mass showed no significant difference comparing with CD group.

Figure. Fat mass (left) and lean mass (right) weights and ratio of the mice by NMR mini Spec LF50. n=4 per group. Differences were assessed by the Student's T-test, *p<0.05; **p<0.01; ***p<0.001; n.s: no significance.

•The studies appear to be performed in relatively young, growing mice. Given they are still in the phase of laying down lean mass, it would seem these studies would be more relevant to the “Catch-up” growth phenomenon put forth by Dulloo et al, rather than weight regain phenomenon commonly seen with dieting in humans. Stunting lean mass growth would change the metabolic requirements of the animals. Figures S1D and S1E are difficult to interpret without body composition data.

Answer:

Thank you for your comments. We have tested the body composition before and after the period of weight regain (Figure shown below). We found that the ratio of the lean mass showed no significant difference before and after weight regain in CR diet group.

Figure. Fat mass (left) and lean mass (right) of mice by miniSpec LF50 NMR. n=4 per group. Differences were assessed by the Student's T-test, *p<0.05; n.s: no significance.

To evaluate the “catch-up” growth phenomenon put forth by Dulloo et al, we observed the 3,5,3'-triiodothyronine (T3) metabolism that plays a major role in the thermogenesis of lean mass. In hindlimb skeletal muscles, T3 metabolism is mainly affected by the iodothyronine deiodinases DIO1, DIO2, and DIO3. Here, we detected the expression levels of these genes after the weight regain period in CR and CD groups and found no significant changes (Figure shown below). We also added this part into the “discussion” part of the manuscript: Ln324-329: “Dulloo et al. found in the rat experiments that the weight regain was mainly contributed by diminished muscle thermogenesis following 50 % CR intervention. In hindlimb skeletal muscles, T3 metabolism is mainly affected by the iodothyronine deiodinases Dio1, Dio2, and Dio3^{25,26}. We found that the expression levels of these genes showed no

significant difference after weight regain period (revised Figure S7E), which might due to the different animal models and different CR degrees and durations.”.

Figure (revised Figure S7E). Gene expression of Dio1, Dio2, Dio3 in hindlimb skeletal muscles of the mice. n=6 per group. Differences were assessed by the Student’s T-test, n.s: no significance.

Additional comments and concerns

•Fig S1B. The authors should justify the diets. They appear to be semi-purified diets which vary in a number of aspects that go beyond the macronutrient composition, as well as the relevance of a 60% fat diet.

Answer: Thank you for your concerns. We used purified diets, including chow diets (TP23522 from Trophic Animal Feed High-tech Co., Ltd), and high-fat diet (D12492 from ResearchDiets, Inc). The ingredients are highly purified according to the diet’s instructions and previous publications ^[1-5]. We have also added the details in the “methods” part (Ln421): “The diets used in this study are highly purified.”.

[1] Alard, J., et al. Beneficial metabolic effects of selected probiotics on diet-induced obesity and insulin resistance in mice are associated with improvement of dysbiotic gut microbiota. *Environ Microbiol*, 18: 1484-97 (2016).

[2] Soto, J E., et al. Comparison of the Effects of High-Fat Diet on Energy Flux in Mice Using Two Multiplexed Metabolic Phenotyping Systems. *Obesity (Silver Spring)*, 27: 793-802 (2019).

[3] Park, Y., et al. Hypoxic exposure can improve blood glycemic control in high-fat diet-induced obese mice. *Physical activity and nutrition*, 24(1), 19–23 (2020).

[4] Zheng, X. , et al. Hyocholic Acid Species Improve Glucose Homeostasis Through a Distinct TGR5 and FXR Signaling Mechanism. *Cell Metabolism*, 33, 4 (2021).

[5] Huang, F., et al. Theabrownin from Pu-erh tea attenuates hypercholesterolemia via modulation of gut microbiota and bile acid metabolism. *Nat Commun* 10, 4971 (2019).

•Ln 30. Thermogenesis was not measured.

Answer: Thanks, and we have added the thermogenesis tests according to your suggestions. The data have been provided in the previous answer of the second “major concerns”, and the revised manuscript.

•Ln 55. The acronym OUT needs to be defined early in the manuscript.

Answer: Thanks, and we have defined the “OTU” at Ln53 complementally: “the operation taxonomic units (OTUs)”.

•Ln 95-98. More increment body weight curves would be more informative.

Answer: Thanks, and we have changed the curves of body weight in the revised Figure1B.

Figure (revised Figure 1B). The body weights of the mice in the fasting experiments. n=5 per group. Differences were assessed by the Student’s T-test, *p<0.05; **p<0.01; ***p<0.001.

•Fig 1A and others. The text is exceptionally small in the workflow figures. Even zooming to 200%, I had trouble reading the text.

Answer: Thanks, and we have made the text font in the workflow figures larger.

•Ln 184-185. The interpretation is overstated- the coincident observations would suggest causality, but they do not provide definitive evidence that the changes were the result of lower CYP7B1 expression.

Answer: Thanks, and we have deleted the causality description of this part.

•Fig S1D. The authors need to clarify how EI was measured and calculated.

Answer: We have changed the figure to “a more refined temporal presentation of EI” based on your advice. We calculated the average daily food energy intake during the 4 weeks of weight regain, and measured the average energy excretion in feces of the mice. We also added the clarification of EI measurements into the manuscript (Ln114-117): “Energy intake is the average value calculated based on the daily food intake of each mouse... No significant changes were found in the excretion of energy in feces...”.

•Ln 203-204. The authors are claiming that the fecal transplant is reducing energy expenditure, but no measures of intake or expenditure are presented to show what might be happening with energy balance.

Answer: Thanks for your suggestion. We have changed the explanation in this part, Ln226: “...After the diet changing experiment, the energy expenditures of the CR+HF group mice were attenuated (revised Figure 1I)...”.

•Fig 1G. Need to clarify the units of the relative gain (presumably g/day).

Answer: We have changed the units of the figure to “Weight gain during 4-weeks HF diet (g)”.

•Ln 218-219. The header should not imply that energy expenditure was changed since it was not measured.

Answer: Thank you for your advice, and we have added the energy expenditure measurements in this part (revised Figure 1I).

Figure (revised Figure 1I). Energy expenditures in the diets changing experiments. n=4 per group. Differences were assessed by the two-way analysis of variance (ANOVA), *p<0.05.

•Ln 247-248. How was energy intake measured and calculated for this experiment? Much of the difference in body weight occurred during the first few days- so the informative measures should come from this early period. Averaging intake over the 30 day follow-up would miss an effect on intake during this early window.

Answer: We agree that much of the difference in body weight occurred during the first few days, so we calculated the average energy intake at the first week during the 4-week weight regain. We have explained this part more clearly in the article Ln266-267: “The average energy intake of each mouse at the first week of recovery...”

•Fig 7B. More incremental measures of EI would be informative, particularly during the window when differences in weight gain occurred. If the data are skewed by measurements toward the end of the study, the differences are likely just a reflection of that body mass differences (metabolic requirements. Again, body composition data would strengthen the studies and their interpretation in support of the fat pad weights presented.

Answer: We calculated the average energy intake at the first week during the 4-week weight regain. We have explained this part more clearly and added the body composition data into the manuscript (revised Figure 6E, 7F).

We revised the manuscript at Ln266-267: “The average energy intake of each mouse at the first week of recovery and the fecal energy excretion among the three groups showed no significant change...”. Ln270: “...decreased weights of the fat mass (revised Figure 6E)”. Ln295: “...the average energy intake of each mouse at the first week of the recovery...”. Ln299: “...decreased weights of fat mass (revised Figure 7F)”.

Figure (revised Figure 6E (left), 7F (right)). Fat mass and lean mass of the mice by miniSpec LF50 NMR. n=4 per group. Differences were assessed by the Student’s T-test, *p<0.05.

•Fig S1F. Morphology from an H&E stain image was used to suggest steatosis—it would be more convincing to have a lipid stain to support this claim.

Answer: Thanks. We have added the lipid stain image into the manuscript (revised Figure S1F).

Figure (revised Figure S1F). The H&E staining of the epididymal white adipose tissue.

•Ln 548-549. Please clarify – you collected blood from the eyeballs? Perhaps you mean from a retro-orbital bleed?

Answer: Thanks. Yes, we collected the blood from the retro-orbital, and we have corrected the description in Ln612: “...we collected blood from the retro-orbital...”

•Ln 395. The authors should consider in the limitations how studying these mice well below thermoneutral conditions may have affected these studies of energy balance, particularly as they are making claims with regard to thermogenesis and energy expenditure.

Answer: Thanks for your comments, we have added the discussion part about this limitation at Ln398-401: “Due to equipment limitations, all the experiments were established at the typical laboratory room temperature (22°C), and the phenomenon of the weight regain model at thermoneutrality (29°C) needs further investigation.”

•Ln 572. It is unclear throughout the paper why the authors jump between parametric to nonparametric tests. Broadly speaking, the whisker box-plots with individual datapoints would be preferred throughout the paper, rather than the bar graphs. The authors should clarify which statistical approach was used for each experiment. In a number of cases, consider presenting absolute values rather than fold change.

Answer: Thanks for your advice, and we have changed the original bar plots into the whisker box plots. We have also unified the statistical methods to parametric tests in the manuscript.

•Figures S6A and s7A appear to be the same with an alternate color scheme. Fig S7A should describe the UCDA experiments.

Answer: Thanks, and we have changed the description in the revised Figure S7A.

Reviewer #3 (Remarks to the Author):

The current submission described a deep and meaningful investigation on the bodyweight rebound after CR treatment. It is an interesting work providing new insights into the mechanism underlying the bodyweight regain after fasting or CR. The experiments design and analysis methods are solid to support the final conclusion. Parabacterial distasonis is the key mediator for the control of body weight. Loss of this gut commensal bacteria after CR is demonstrated to be the causes for bodyweight regaining. And also, alteration of non12-OH bile due to the change of gut microbiome, especially the decrease of P. distasonis, is confirmed as the gut metabolites mediating the recovery effects. This work is well organized and clearly written.

To my judgement, it can be accepted in its current version.

Answer: Thank you for your positive comments.

REVIEWER COMMENTS

Reviewer #1 (Remarks to the Author):

The authors have addressed all my comments and are to be commended for their efforts in establishing causality for many aspects of the proposed mechanism that were only associatively linked in the previous version of the manuscript.

The only (stylistic) remark I have is that Figure 1H and Supplementary Fig. S1H now largely show the same data. It would make more sense to replace Figure 1H with the graph that is now in Supplementary Fig. S1H.

Reviewer #2 (Remarks to the Author):

The authors have examined changes in the microbiome, bile acids, and gene expression in models of weight regain. HF feeding and calorie restriction ((CR) led to similar changes in the microbiome and BA metabolite profiles in the gut and serum. A decline in *Parabacteroides distasonis* (PD) was common for both, and transplant and supplement studies with PD or the non-12OH bile acids attenuated weight regain. Overall, the authors present evidence of a role of the gut microflora in mediating some aspects of the biological drive to regain weight, and they provide evidence for a mechanism that engages PD, bile acids, and their effect on gene expression in peripheral tissues. The revised manuscript includes a number of additional studies, most prominent of which are measures of energy expenditure and body composition that support the thesis. With the new data provided in the revised manuscript, I have some additional issues for the author to consider.

Major Comments

- I am still not clear as to what the “fasting” paradigm is modeling. The step-down or step-up isn’t well justified, and it should be noted that 4 days without food for a mouse is an exceptionally extreme metabolic stress, akin to prolonged starvation. Most laboratory mice would fail to survive with 4 or 5 days without food. Regardless, the minimal loss in lean body mass is surprising given the extreme intervention. The 50%CR pre and post the 4 days of fasting seem to complicate what the authors are looking at.
- The authors should be commended for bring a wealth of metabolic data to the revised manuscript, as it provides important information to support the authors’ arguments. The energy expenditure data, however, need to be expressed in absolute (without normalization, kcal/day) in relation to food intake (kcal/day), as these actual parameters are what directly inform energy balance. The authors also need to present the body composition data that is being used to normalize the metabolic data in Figure 1I (lean mass, fat mass). The normalization to lean mass as presented is important and relevant to the question, but it only serves as an explanation for what might be happening with energy balance.
- Actual serum concentrations of LCA and UDCA should be presented, in addition to total serum BA and bile acid % composition analysis. This is important because the authors claim that much of their phenotype is due to reductions in LCA or UDCA signaling through Tgr5. In Figure 3, the authors show

substantially-increased total BAs in the CR mice. Even though LCA and UDCA were decreased as a percentage of the total BA pool in CR mice, was their final serum concentrations of these species lower than those of the CD and HF groups? The y-axis of their serum bile acid composition graphs would be better labelled as 'percent composition' rather than 'concentration'.

Other comments

- Ln 119-120. The authors need to clarify that... "energy expenditure, as normalized to lean mass, was significantly reduced. Unless the authors show the absolute (raw, unnormalized) energy expenditure data showing that same difference. The distinction here is important in that often is it the difference on body weight or lean mass that drives the difference.
- The promethion metabolic system has the capability of providing spontaneous physical activity, so it is unclear why this is not reported. Even if there were no differences, it would serve the purpose of ruling this out as an explanation for the differences in energy expenditure.
- More parameters for the UCP1 knockout (UKO) mice would have been helpful. What was their food intake, fasting glucose, GTT, etc. Did the microbiome and BA composition change the same way in UKO mice as they change in the WT C57?
- The authors looked at DIO1, DIO2, and DIO3 gene expression in the skeletal muscle with no difference, but what about brown adipose tissue? This is a well known target of Tgr5 signaling, and DIO2 is also part of the thermogenic program in brown fat? Since DIO2 is a key target of Tgr5, I am curious why the authors didn't see this gene change in skeletal muscle.
- While the authors have presented an exceptional amount of work, a nice complimentary experiment would have been for the authors to use a highly potent synthetic Tgr5 agonist such as INT-777 (if available).
- The relevance of the 60% HFD remains questionable.
- The authors are still pushing the limits on claiming causality, particularly with the Tgr5 signaling. Other potential mechanisms of increased thermogenic program upregulation (e.g. FGF21, irisin, increased SNS, etc) remain possibilities.

REVIEWER COMMENTS

Reviewer #1 (Remarks to the Author):

The authors have addressed all my comments and are to be commended for their efforts in establishing causality for many aspects of the proposed mechanism that were only associatively linked in the previous version of the manuscript.

The only (stylistic) remark I have is that Figure 1H and Supplementary Fig. S1H now largely show the same data. It would make more sense to replace Figure 1H with the graph that is now in Supplementary Fig. S1H.

Answer: Thanks for your comments, we have changed the Figure 1H accordingly (see below).

Figure 1H. Bodyweights during the experiment, the differences were compared with the CD+HD group.

Reviewer #2 (Remarks to the Author):

The authors have examined changes in the microbiome, bile acids, and gene expression in models of weight regain. HF feeding and calorie restriction ((CR) led to similar changes in the microbiome and BA metabolite profiles in the gut and serum. A decline in Parabacteroides distasonis (PD) was common for both, and transplant and supplement studies with PD or the non-12OH bile acids attenuated weight regain. Overall, the authors present evidence of a role of the gut microflora in mediating some aspects of the biological drive to regain weight, and they provide evidence for a mechanism that engages PD, bile acids, and their effect on gene expression in peripheral tissues. The revised manuscript includes a number of additional studies, most prominent of which are measures of energy expenditure and body composition that support the thesis. With the new data provided in the revised manuscript, I have some additional issues for the author to consider.

Major Comments

- I am still not clear as to what the “fasting” paradigm is modeling. The step-down or step-up isn’t well justified, and it should be noted that 4 days without food for a mouse is an

exceptionally extreme metabolic stress, akin to prolonged starvation. Most laboratory mice would fail to survive with 4 or 5 days without food. Regardless, the minimal loss in lean body mass is surprising given the extreme intervention. The 50%CR pre and post the 4 days of fasting seem to complicate what the authors are looking at.

Answer: Thank you for the questions regarding our design of the fasting experiment. In fact, we had also extensively discussed these concerns prior to the experiment.

The study presented in our paper was carefully designed and technically justified. All the procedures regarding animal maintenance and intervention are in strict accordance with the policy of the Institutional Animal Care and Use Committee (IACUC) of Shanghai Jiao Tong University Affiliated Sixth People's Hospital. In the study, we took into careful consideration of the mouse physical status. Firstly, all of the experimental mice had free access to drinking water during diet restriction and fasting. Mice were individually housed in clean cages to reduce cannibalism, coprophagy, and residual chow. Secondly, we gradually reduced the food intake from free access to food, to a 4-day 50% caloric restriction, and then to a 4-day fasting, to allow their metabolic and physiological adaptation to complete food deprivation. This step-down or step-up design has precedents, the purpose is to reduce deaths caused by sudden and prolonged fasting¹. Thirdly, during the entire diet restriction period, the physical status was closely observed (three times a day). Mice were humanely euthanized when the weight loss in one day was greater than 8%. Moreover, the duration of the 4-day fasting was determined based on technical discussions, consultation with the attending veterinarian of the animal facility, as well as a thorough review of previously published mouse studies which lasted 4 days¹⁻⁴. We want to figure out how the intestinal flora can be altered to the greatest extent (within the achievable range) in fasting.

Once again, thank you for your thoughts and concerns on the study.

[1] Zheng X, et al. Food withdrawal alters the gut microbiota and metabolome in mice. *FASEB J.* 2018 Sep;32(9):4878-4888.

[2] Kaneda T, et al. Hashimoto K. Differential neuropeptide responses to starvation with ageing. *J Neuroendocrinol.* 2001 Dec;13(12):1066-75.

[3] Clausen J, Konat G. Enzymic and behavioural changes in mice fed polychlorinated biocides followed by starvation. *Experientia.* 1972 Aug 15;28(8):902-3.

[4] Byrne BM, Dankert J. Volatile fatty acids and aerobic flora in the gastrointestinal tract of mice under various conditions. *Infect Immun.* 1979 Mar;23(3):559-63.

•The authors should be commended for bring a wealth of metabolic data to the revised manuscript, as it provides important information to support the authors' arguments. The energy expenditure data, however, need to be expressed in absolute (without normalization, kcal/day) in relation to food intake (kcal/day), as these actual parameters are what directly inform energy balance. The authors also need to present the body composition data that is being used to normalize the metabolic data in Figure 1I (lean mass, fat mass). The normalization to lean mass as presented is important and relevant to the question, but it only serves as an explanation for what might be happening with energy balance.

Answer: Thank you for your comments. We agree that the absolute value of the metabolic data show directly the energy balance information, thus the metabolic readouts without

normalization are shown in the revision. In consideration of the nocturnal lifestyle of mice, we calculated the average energy expenditures during the night and during the light respectively. As to the p value calculation, we referred to the newest papers about energy data analysis ¹, testing the p value of analysis-of-covariance (ANCOVA), with body mass as a covariate (CD+HF vs CR+HF: p=0.0026).

The revised paragraph was (Ln123-127): “The energy expenditure of CR+HF mice was significantly lower than that of control mice in the dark stage, which supports the hypothesis that weight regain occurred by decreasing energy expenditure (Figure 1H).”

Figure 1H. The energy expenditure in three groups. Differences were assessed by the Student's T-test, *p<0.05.

[1] Müller TD, et al. Revisiting energy expenditure: how to correct mouse metabolic rate for body mass. Nat Metab. 2021 Sep;3(9):1134-1136.

•Actual serum concentrations of LCA and UDCA should be presented, in addition to total serum BA and bile acid % composition analysis. This is important because the authors claim that much of their phenotype is due to reductions in LCA or UDCA signaling through Tgr5. In Figure 3, the authors show substantially-increased total BAs in the CR mice. Even though LCA and UDCA were decreased as a percentage of the total BA pool in CR mice, was their final serum concentrations of these species lower than those of the CD and HF groups? The y-axis of their serum bile acid composition graphs would be better labelled as 'percent composition' rather than 'concentration'.

Answer: Thanks for your comments in the important point, and we have taken this into consideration during our data exploration. Here, we represented the absolute values of the bile acids (BAs) in CD, HF and CR groups (figure see below, Figure R1). We agree that the final serum concentration of UDCA was increased in the CR group as the synthesis of total BAs and BAs in serum were 4 times increased in the CR group, LCA showed no significant changes. In addition to non-12OH BAs (such as UDCA), many 12-OH BAs were also increased to a larger proportion (such as CA and DCA). We believe that the BA pool is very complex and individual BAs impact the BA receptor signaling pathways *via* their ability to act as either agonists or antagonists. Therefore, the BA composition is just as important as the absolute concentration of any BA. We tested for the TGR5 gene expression, which was shown to be

decreased in the CR group (see figure below, Figure R2). More and more studies use the percentage data in BAs analysis¹⁻⁵. We believe that the percentage data will show the more comprehensive results in BAs integrated functions. Moreover, we have revised the label in the y-axis as “concentration percentage (%)”, and the descriptions in the manuscript according to your suggestion.

Figure R1. The concentration of BAs in serum. N=8, differences were assessed by the Student’s T-test, *p<0.05.

Figure R2. The mRNA expressions of TGR5 in CD, HF and CR group. N=8, differences were assessed by the Student’s T-test.

- [1] Jiao N, et al. Suppressed hepatic bile acid signalling despite elevated production of primary and secondary bile acids in NAFLD. *Gut*. 2018 Oct;67(10):1881-1891.
- [2] Caussy C, et al. Serum bile acid patterns are associated with the presence of NAFLD in twins, and dose-dependent changes with increase in fibrosis stage in patients with biopsy-proven NAFLD. *Aliment Pharmacol Ther*. 2019 Jan;49(2):183-193.
- [3] Duboc H, et al. Connecting dysbiosis, bile-acid dysmetabolism and gut inflammation in inflammatory bowel diseases. *Gut*. 2013 Apr;62(4):531-9.

[4] Bamba S, et al. Relationship between the gut microbiota and bile acid composition in the ileal mucosa of Crohn's disease. *Intest Res.* 2021 May 14.

[5] Stepien M, et al. Pre-diagnostic alterations in circulating bile acid profiles in the development of hepatocellular carcinoma. *Int J Cancer.* 2021 Nov 29. Epub ahead of print.

Other comments

•Ln 119-120. The authors need to clarify that... “energy expenditure, as normalized to lean mass, was significantly reduced. Unless the authors show the absolute (raw, unnormalized) energy expenditure data showing that same difference. The distinction here is important in that often is it the difference on body weight or lean mass that drives the difference.

Answer: Thank you for your advice, we showed the raw data in the revised paper. The revised paragraph was (Ln123-127): “The energy expenditure of CR+HF mice was significantly lower than that of control mice in the dark stage, which supports the hypothesis that weight regain occurred by decreasing energy expenditure (Figure 1H). The decreasing energy expenditure was found to be independent of physical activity (ANOVA $p=0.39$, Figure S1F).”.

•The prometheon metabolic system has the capability of providing spontaneous physical activity, so it is unclear why this is not reported. Even if there were no differences, it would serve the purpose of ruling this out as an explanation for the differences in energy expenditure.

Answer: Thank you for your advice, we added the physical activity data to the revised paper. The revised paragraph is:

Ln125: “The decreasing energy expenditure was found to be independent of physical activity (ANOVA $p=0.39$, Figure S1F).”.

Ln276: “The intervention of PD...and the physical activity showed no significant difference (ANOVA $p=0.19$, Figure S6E).”.

Ln308: “The administration of the non-12OH BA ..., which was independent of the physical activity (ANOVA $p=0.17$, Figure S7D).”

Figure S1F. The locomotor activity in three groups. $n=4$ per group. Differences were assessed by the two-way analysis of variance (ANOVA), $p=0.39$.

Figure S6E. The locomotor activity in three groups. n=4 per group. Differences were assessed by the two-way analysis of variance (ANOVA), $p=0.19$.

Figure S7D. The locomotor activity in three groups. n=4 per group. Differences were assessed by the two-way analysis of variance (ANOVA), $p=0.17$.

•More parameters for the UCP1 knockout (UKO) mice would have been helpful. What was their food intake, fasting glucose, GTT, etc. Did the microbiome and BA composition change the same way in UKO mice as they change in the WT C57?

Answer: Thanks for your valuable advice. We tested the food intake, fasting glucose, GTT, and ITT (Figure R3). Additionally, we tested BA composition change (Figure R4) as well as the microbial metabolites (Figure R5) in UKO mice, which showed the same variation tendency with WT C57 mice.

We revised the manuscript to:

Ln291-294: "In UCP1-knockout (UKO) mice, the intervention of PD shifted the microbial metabolites towards what was found in the wild-type mice (Figure S6J) and elevated the proportion of non-12OH BAs (Figure S6K, L) but could not rescue the weight regain (Figure 6O), as well as the fasting blood glucose (Figure 6P) and AUCs in OGTT and ITT tests (Figure S6M)."

Ln317:321: "In UKO model, elevated non-12OH BAs (Figure S6K, L) could not significantly reduce the body weights (Figure 7J) and the fasting glucose (Figure 7K) and AUC of ITT test in CR+HF mice (Figure S6M). Interestingly, we found the AUC of OGTT decreased in CR+HF+UDCA UKO mice, implying the influence of BAs on other pathways of blood glucose regulation."

Figure R3. The average food intake (A) and fasting blood glucose (B), OGTT (C) and ITT (D) glucose levels and AUC in UKO mice. Differences were assessed by the Student's T-test, n.s: no significance, * $p < 0.05$.

Figure R4 (Figure S6K, L). The percentage of 12OH BAs /non-12OH BAs (left), LCA, and UDCA (right) in serum. Differences were assessed by the Student's T-test, n.s: no significance, ** $p < 0.01$.

Figure R5 (Figure S6J). The PLSDA plot (left) and the variate 2 (right, the variate of y-axis in the PLSDA plot) of microbial metabolites profiling in WT and UKO mice serum. Differences were assessed by the Student's T-test, n.s: no significance.

•The authors looked at DIO1, DIO2, and DIO3 gene expression in the skeletal muscle with no difference, but what about brown adipose tissue? This is a well known target of Tgr5 signaling, and DIO2 is also part of the thermogenic program in brown fat? Since DIO2 is a key target of Tgr5, I am curious why the authors didn't see this gene change in skeletal muscle. Answer: Thanks for your valuable advice. We tested the Dio2 mRNA expression in BAT, which showed a decreased in the CR+HF group mice (Figure R4). But there was no significant change in skeletal muscle. We need to develop a more in-depth study of the metabolic

changes in muscle (including other thermogenesis-related genes). Thank you for the valuable enlightenment.

Figure R4. The mRNA expressions of Dio2 in BAT. N=8, differences were assessed by the Student's T-test.

- While the authors have presented an exceptional amount of work, a nice complimentary experiment would have been for the authors to use a highly potent synthetic Tgr5 agonist such as INT-777 (if available).

Answer: Thank you for your advice. Yes, Tgr5 agonists have been shown to prevent obesity and insulin resistance in many mouse studies ¹⁻³. Based on these studies, we hypothesized that a Tgr5 agonist (such as INT-777) would alleviate weight regain. Work is in progress in our lab on the function of TGR5 in insulin resistance. Thank you for your comments.

[1] de Oliveira MC, et al. Bile acid receptor agonists INT747 and INT777 decrease oestrogen deficiency-related postmenopausal obesity and hepatic steatosis in mice. *Biochim Biophys Acta*. 2016 Nov;1862(11):2054-2062.

[2] Wang XX, et al. FXR/TGR5 Dual Agonist Prevents Progression of Nephropathy in Diabetes and Obesity. *J Am Soc Nephrol*. 2018 Jan;29(1):118-137.

[3] Ding L, et al. Notoginsenoside Ft1 acts as a TGR5 agonist but FXR antagonist to alleviate high fat diet-induced obesity and insulin resistance in mice. *Acta Pharm Sin B*. 2021 Jun;11(6):1541-1554.

- The relevance of the 60% HFD remains questionable.

Answer: This 60% HFD diet is the purified ingredient diet and not a grain-based diet, which has 60% kcal energy from fat. In the purified ingredient diets. The differences between groups could thus be due to the level of fat. It is a kind of “original” high-fat diets for diet induced obesity. We also explained the diets in the manuscript in more detail (Ln438-439): “The diets used in this study are purified ingredient diets and not the grain-based diets, differences between groups could be related to the level of fat...”.

- The authors are still pushing the limits on claiming causality, particularly with the Tgr5 signaling. Other potential mechanisms of increased thermogenic program upregulation (e.g. FGF21, irisin, increased SNS, etc) remain possibilities.

Answer: Thank you for your comprehensive comments. We agreed that Tgr5 signaling is a key signaling but may not be the only signaling in metabolic changes, and other potential

mechanisms of increased thermogenic program upregulation (e.g. FGF21, irisin, increased SNS, etc) remain possibilities. So, we added the description in the discussion part (Ln408-Ln411): “Here, we saw the Tgr5 expression decreased in the CR+HF group, which can lead to metabolic changes that increase weight regain. Other potential mechanisms of thermogenic program upregulation may provide additional therapeutic options (e.g. Fgf21, irisin, increased SNS) and require further investigations.”.

REVIEWER COMMENTS

Reviewer #2 (Remarks to the Author):

The authors adequately addressed a number of my concerns. I have two remaining follow-up points that the authors need to address.

1. The authors should be commended for now presenting the raw energy expenditure (Figure 1H), as it is these values that have direct relevance for the energy balance equation. It would be pertinent to do the appropriate statistical comparison on these raw values and indicate whether this is actually suppressed with the treatment or if there is a trend. Then, if the authors would also like to adjust the data (for body mass or the more relevant parameter of lean mass) using ANCOVA as they state, that is appropriate as long as the two analyses (raw vs adjusted) are not conflated or confused. My impression with the revision is that the left panel is the raw data expressed as metabolic rate per hour over the day, and the right panel are ANCOVA adjusted data. If this is the case, the right panel needs to indicate the latter is “adjusted” energy expenditure and clarify the distinction in the y axis of the panel and specify the analysis with ANCOVA and the adjustment from body weight in the figure legend. My broader point here is that in the text, figures and interpretation, it is important to make the distinction between “energy expenditure” and “energy expenditure adjusted for body mass”. These are two distinct parameters with important and distinct implications for the reader to appreciate.

2. The authors could be correct that “percent composition” is the more important expression of this and other bile acid analyses in Figure 3, rather than absolute values, but I would disagree that this is a consensus opinion and that better transparency is warranted. The authors should reference and make Figure 1R in the rebuttal available to the reader, at least in the supplemental data, and perhaps provide the argument for the relevance of percent composition in the text for the reader to consider. This is particularly important in this case in that there is a decrease percent composition in the potent circulating TGR5 agonists LCA and UDCA, but the actual circulating concentrations of these agonists were unchanged or even elevated in the CR mice compared to CD. Because of this discrepancy, we would encourage the authors to provide both percent composition and absolute concentrations, and provide the salient arguments for the reader as to why the former is the most relevant in this case.

REVIEWER COMMENTS

Reviewer #2 (Remarks to the Author):

The authors adequately addressed a number of my concerns. I have two remaining follow-up points that the authors need to address.

1. The authors should be commended for now presenting the raw energy expenditure (Figure 1H), as it is these values that have direct relevance for the energy balance equation. It would be pertinent to do the appropriate statistical comparison on these raw values and indicate whether this is actually suppressed with the treatment or if there is a trend. Then, if the authors would also like to adjust the data (for body mass or the more relevant parameter of lean mass) using ANCOVA as they state, that is appropriate as long as the two analyses (raw vs adjusted) are not conflated or confused. My impression with the revision is that the left panel is the raw data expressed as metabolic rate per hour over the day, and the right panel are ANCOVA adjusted data. If this is the case, the right panel needs to indicate the latter is “adjusted” energy expenditure and clarify the distinction in the y axis of the panel and specify the analysis with ANCOVA and the adjustment from bodyweight in the figure legend. My broader point here is that in the text, figures and interpretation, it is important to make the distinction between “energy expenditure” and “energy expenditure adjusted for body mass”. These are two distinct parameters with important and distinct implications for the reader to appreciate.

Answer: Thank you for the advice. The energy expenditure data provided in Figure 1H (both in the left and right panel) are the raw data without adjustment, the units of which are Kcal/h. We apologize that we didn't make it clear in the figure legend. The average energy expenditures per hour in 24 hours were shown in the left panel, and the average energy expenditures per hour during the period of light (8:00 a.m.-8:00 p.m.) and during the period of dark (8:00 p.m.-8:00 a.m. next day) were shown in the right panel. The statistical comparison was assessed by the Student's T-test (right panel), and the results showed that the energy expenditures were significantly suppressed with CR intervention during the period of darkness (* $p < 0.05$). Due to the methodological pitfalls that reside in mass-specific metabolic rate indices, previous studies conducted by other researchers suggested to abandon ratio-based measures of metabolic rate in favor of regression-based analysis-of-covariance (ANCOVA), with body mass as a covariate^[1,2]. Thus, we also applied the statistical comparison ANCOVA with body mass as a covariate, and the results showed that the energy expenditures were significantly suppressed across 24-h time with body mass adjustment (# $p < 0.05$).

We revised the figure legend: “H) Raw energy expenditures per hour in 24 hours (left) and raw average energy expenditures per hour during the period of light and dark(right). Differences in the left panel were assessed by the analysis-of-covariance (ANCOVA) with body mass as a covariate, # $p < 0.05$. Differences in the right panel were assessed by the Student's T-test, * $p < 0.05$.”.

Figure 1H. Raw energy expenditures per hour in 24 hours (left) and raw average energy expenditures per hour during the period of light and dark(right). Differences in the left panel were assessed by the analysis-of-covariance (ANCOVA) with body mass as a covariate, # $p < 0.05$. Differences in the right panel were assessed by the Student's T-test, * $p < 0.05$.

[1] Tschöp MH, et al. A guide to analysis of mouse energy metabolism. *Nat Methods*. 2011 Dec 28;9(1):57-63.

[2] Müller TD, et al. Revisiting energy expenditure: how to correct mouse metabolic rate for body mass. *Nat Metab*. 2021 Sep;3(9):1134-1136.

2. The authors could be correct that “percent composition” is the more important expression of this and other bile acid analyses in Figure 3, rather than absolute values, but I would disagree that this is a consensus opinion and that better transparency is warranted. The authors should reference and make Figure 1R in the rebuttal available to the reader, at least in the supplemental data, and perhaps provide the argument for the relevance of percent composition in the text for the reader to consider. This is particularly important in this case in that there is a decrease percent composition in the potent circulating TGR5 agonists LCA and UDCA, but the actual circulating concentrations of these agonists were unchanged or even elevated in the CR mice compared to CD. Because of this discrepancy, we would encourage the authors to provide both percent composition and absolute concentrations, and the provide the salient arguments for the reader as to why the former is the most relevant in this case.

Answer: Thank you for your comments and advice. We have provided the relevant absolute values of bile acids in the supplemental information, and the discussion parts in the manuscript.

The manuscript has been revised at Ln189/Ln226/ Ln245: “the relevant absolute values are showed in Figure S3B/S4E/S5D”.

Supplementary Fig. S3B, S4E, S5D. The absolute concentration of BAs in serum. Differences with CD or CD+HF group were assessed by the Student's T-test, *p<0.05.

We have also revised the discussion (Ln412-Ln420): “It should be noted that due to the increased synthesis of total BA after CR, many 12OH BAs (such as CAs and DCAs) were increased to a larger proportion (Figure S3B). However, the absolute concentrations of non-12OH BAs (such as UDCA) remained unchanged. We believe that the BA pool is influenced by many factors and that individual BAs impact the BA receptor signaling pathways via their ability to act as either agonists or antagonists. Therefore, the form of BA percentage is increasingly used in BA data analysis and presentation [1-5], as in this study, providing more comprehensive results to display integrated biological functions of BAs.”

[1] Jiao N, et al. Suppressed hepatic bile acid signalling despite elevated production of primary and secondary bile acids in NAFLD. *Gut*. 2018 Oct;67(10):1881-1891.

[2] Caussy C, et al. Serum bile acid patterns are associated with the presence of NAFLD in twins, and dose-dependent changes with increase in fibrosis stage in patients with biopsy-proven NAFLD. *Aliment Pharmacol Ther*. 2019 Jan;49(2):183-193.

[3] Duboc H, et al. Connecting dysbiosis, bile-acid dysmetabolism and gut inflammation in inflammatory bowel diseases. *Gut*. 2013 Apr;62(4):531-9.

[4] Bamba S, et al. Relationship between the gut microbiota and bile acid composition in the ileal mucosa of Crohn's disease. *Intest Res*. 2021 May 14.

[5] Stepien M, et al. Pre-diagnostic alterations in circulating bile acid profiles in the development of hepatocellular carcinoma. *Int J Cancer*. 2021 Nov 29. Epub ahead of print.

REVIEWERS' COMMENTS

Reviewer #2 (Remarks to the Author):

The authors have addressed my concerns.

REVIEWERS' COMMENTS

Reviewer #2 (Remarks to the Author):

The authors have addressed my concerns.

Answer: Thank you for your valuable comments on our revised manuscript.